# FAME: Adaptive Functional Attention with Expert Routing for Function-on-Function Regression

**Yifei Gao**
Department of Industrial Engineering
Tsinghua University
Beijing 100084, China
gao-yf@mail.tsinghua.edu.cn

**Yong Chen**
Department of Industrial and Systems Engineering
University of Iowa
Iowa City, IA 52242, USA
yong-chen@uiowa.edu

**Chen Zhang**[*]
Department of Industrial Engineering
Tsinghua University
Beijing 100084, China
zhangchen01@tsinghua.edu.cn

## Abstract

Functional data play a pivotal role across science and engineering, yet their infinite-dimensional nature makes representation learning challenging. Conventional statistical models depend on pre-chosen basis expansions or kernels, limiting the flexibility of data-driven discovery, while many deep-learning pipelines treat functions as fixed-grid vectors, ignoring inherent continuity. In this paper, we introduce **F**unctional **A**ttention with a **M**ixture-of-**E**xperts (FAME), an end-to-end, fully data-driven framework for function-on-function regression. FAME forms continuous attention by coupling a bidirectional neural controlled differential equation with MoE-driven vector fields to capture intra-functional continuity, and further fuses change to inter-functional dependencies via multi-head cross attention. Extensive experiments on synthetic and real-world functional regression benchmarks show that FAME achieves state-of-the-art accuracy and strong robustness to arbitrarily sampled discrete observations of functions.

## 1 Introduction

Functional data are samples whose individual elements are random continuous functions, providing rich temporal–spatial dynamics in domains such as biology [13], marketing [11], transportation [17], and meteorology [42]. With the growing demand for high-resolution measurements, functional data analysis (FDA) is attracting increasing attention in both scientific research and practical applications [41, 27, 20, 48]. Among canonical problems in functional data analysis (FDA), **function-on-function regression (FoFR)** is widely regarded as the most challenging and representative task, as it requires the model to accommodate both infinite-dimensional inputs and outputs. [35, 25, 9]. While functional data offer new opportunities, they also bring intertwined challenges: **(1) Intra-functional continuity**: each function lies in infinite-dimensional continuous functional space with features such as local dynamics and global trends [29]; **(2) Inter-functional interactions**: different dimensions of the input function can have nonlinear couplings with each other [6]; **(3) Feature heterogeneity**: different functions may exhibit vastly different properties such as scale, smoothness, or noise, and may even reside in different functional spaces [7].

---

[*]Corresponding author

39th Conference on Neural Information Processing Systems (NeurIPS 2025).

To address these challenges, existing approaches broadly fall into two methodological categories. The first category comprises classical statistical methods, which manage functional complexity by projecting each function onto finite-dimensional representations. Common techniques include basis expansions, such as B-splines [5], wavelets [49], and functional principal component analysis (FPCA) [8], as well as kernel-based methods that embed functions into functional reproducing kernel Hilbert spaces via operator-valued kernels [4]. Although these methods facilitate linear or additive modeling in lower-dimensional spaces, their performance heavily depends on predefined bases or kernels and strong smoothness assumptions, limiting their ability to capture intra-functional and inter-functional features prevalent in real-world functional data. The second category involves recent deep learning approaches, which offer greater flexibility inspired by advances in natural language processing and computer vision. These methods typically feed discretized curves into deep neural networks. For instance, Shi et al. [33] proposes a smooth-kernel neural network tailored to nonlinear functional regression. Yao et al. [45] introduces adaptive basis learners that adjust expansions according to specific tasks. Although such models enhance nonlinearity and support end-to-end learning, they still assume all functions lie in discretized regular sampling grids with the same dimension, and treat functional data as finite-dimensional vectors. Consequently, these methods have limited power to model intra-functional continuous dynamics, and cannot be applied to irregularly sampled cases, let alone handle feature heterogeneity.

To overcome these limitations, we introduce Functional Attention with Mixture-of-Experts (FAME), an end-to-end framework designed expressly for FOFR. Specifically, our contributions are: (1) FAME is the first FoFR model that directly operates on irregularly sampled functional space without relying on predefined basis functions or discretisation grids. Its effectiveness is rigorously established through both theoretical guarantees and extensive experimental validation. (2) FAME includes a novel functional attention mechanism composed of continuous attention via bidirectional Neural controlled differential equations (NCDEs), which can efficiently capture intra-functional continuity, and multi-head cross attention, which can efficiently model inter-functional interactions. (3) FAME enhances a mixture-of-experts (MoE) architecture, enabling adaptive modeling of feature heterogeneity, and incorporates an NCDE decoder capable of generating continuous functional outputs at arbitrary query locations, naturally accommodating misaligned target indices.

## 2   Related work

**Function-on-function regression**   FoFR has witnessed substantial advances in recent years and is drawing increasing attention from the research community. Existing approaches can be concretely divided into three categories: linear decomposition, reproducing kernel Hilbert space (RKHS), and deep learning methods. Linear approaches employ finite bases such as FPCA, wavelets, or splines to project input and response curves into low-dimensional coordinates, enabling conventional regression techniques [44, 24, 23, 22, 28]. Despite computational efficiency, these approaches rely on fixed bases, restricting their ability to capture non-stationary dynamics and complex interactions. RKHS-based methods use operator-valued kernels, providing nonlinear function regression within linear frameworks [19, 14]. Extensions have incorporated robust losses [16] and optimal estimation strategies [35], yet kernel selection and scalability remain persistent challenges. Deep learning methods, including functional neural networks (FNNs) [30], FPCA-based neural networks [39, 40], and adaptive basis expansions [45], offer data-driven flexibility. However, these techniques typically discretize functions onto fixed grids, ignoring the inherent continuity and heterogeneity of functional data. In summary, current methods rely on either predefined or prematurely discretized feature representation for functional data, limiting their ability to model complex FoFR problems comprehensively.

**Neural differential equations**   Neural differential equations provide a powerful framework that combines continuous-time dynamics with the high-capacity function approximation of neural networks, making them particularly effective for handling irregular time series [31, 12]. Among these methods, Neural ODEs map discrete data to continuous trajectories and leverage adjoint methods for memory-efficient gradient computation, enabling scalable training for large models [1]. Building on this concept, Neural CDEs [15] extend the approach by capturing the continuous-time dynamics of RNNs through a learned control function for hidden states. This strategy has shown superior performance in modeling irregular time series compared to Neural ODEs or RNNs, particularly in offline prediction tasks [3]. Recent work also emphasizes their usefulness in enhancing stability

and generalization by injecting noise directly into the training process [26]. However, Neural CDEs operate in a strictly causal fashion, building hidden-state dynamics solely from past observations, whereas FoFR can exploit the full input function to capture global dependencies and non-causal relationships.

**Attention mechanism for feature representation** The attention mechanism was initially proposed to mitigate the limitations of encoding sequences into fixed-length vectors in RNN-based models [2]. Xu et al. [43] introduces soft alignment during decoding, enabling better handling of long sequences. This concept evolves through variants like global and local attention, and reaches a milestone with the Transformer architecture [37], which entirely replaces recurrence with self-attention, improving both modeling power and computational efficiency. Building on this foundation, numerous attention-based models have emerged for time series modeling. For example, iTransformer[21] reorganises the input so that time steps serve as channels and variables act as tokens, allowing self-attention to capture long-range temporal dependencies more effectively, whereas CrossFormer[47] employs a frequency-aware, cross-scale attention mechanism to model long- and short-term patterns simultaneously. However, these attention mechanisms are designed for discrete sequences defined on fixed grids. They do not provide a native framework for continuous-space mapping of functions.

## 3 Function-on-function regression

We consider an unknown operator $\mathcal{T} : \mathcal{X} \to \mathcal{Y}$ that maps a $d$-tuple of input functions $X = (X^{(1)}, \ldots, X^{(d)})$ to an $m$-tuple of output functions $Y = (Y^{(1)}, \ldots, Y^{(m)})$. For every input channel $j \in \{1, \ldots, d\}$ the function $X^{(j)} : [0, T_j] \to \mathbb{R}$ is assumed to have finite 1-variation; we denote the space by

$$\mathcal{V}_j = \big\{ X^{(j)} : [0, T_j] \to \mathbb{R} \mid \|X^{(j)}\|_{1-\mathrm{var}} < \infty \big\}, \quad \mathcal{X} = \mathcal{V}_1 \times \cdots \times \mathcal{V}_d.$$

For each output index $\zeta \in \{1, \ldots, m\}$, the output function $Y^{(\zeta)} : [0, S_\zeta] \to \mathbb{R}$ is continuous and equipped with the supremum norm $\|Y^{(\zeta)}\|_\infty = \sup_{s \in [0, S_\zeta]} |Y^{(\zeta)}(s)|$. We set

$$\mathcal{W}_\zeta = C\big([0, S_\zeta], \mathbb{R}\big), \quad \mathcal{Y} = \mathcal{W}_1 \times \cdots \times \mathcal{W}_m, \quad \|Y\|_\infty = \max_\zeta \|Y^{(\zeta)}\|_\infty.$$

In practice, for each sample $i$ we observe $X_i = (X_i^{(1)}, \ldots, X_i^{(d)}) \in \mathcal{X}, \quad Y_i = (Y_i^{(1)}, \ldots, Y_i^{(m)}) \in \mathcal{Y}$, but only at irregular time points $\{(t_{i,\ell}, x_{i,\ell})\}_{\ell=1}^{\Lambda_i}$ for each $X_i^{(j)}$ and $\{(s_{i,r}, y_{i,r})\}_{r=1}^{\Gamma_i}$ for each $Y_i^{(\zeta)}$. Given a dataset $\{(X_i, Y_i)\}_{i=1}^{N_s} \subset \mathcal{X} \times \mathcal{Y}$, our goal is to learn a finite-parameter approximation $\mathcal{T}_\theta$ that recovers $\mathcal{T}$ from these discretely and non-uniformly observed functions.

## 4 Functional attention with a mixture-of-experts (FAME)

In this section, we describe how FAME addresses the core challenges of FoFR by designing a custom functional attention mechanism. Figure 1 presents a high-level overview of the overall architecture of FAME. To account for the truly infinite-dimensional nature of each function $X^{(j)}$, we introduce in Section 4.1 a continuous attention block built on a bidirectional NCDE that integrates both forward and backward over $t$. This construction produces smoothly varying Query, Key, and Value trajectories, ensuring that every query instant attends to both past and future contexts and remains insensitive to the discrete sampling along the input function. To capture heterogeneity across functions, Section 4.2 enhances this continuous attention block by replacing its single vector field with a mixture of $K$ expert fields (MoE), each specialised for distinct scales, amplitudes, or noise regimes. A learnable router allocates a weight vector over the $K$ experts, thereby boosting the model's ability to capture feature heterogeneity, removing the need to instantiate a separate encoder for each function, and enabling smooth generalisation to unseen features. Section 4.3 then introduces a cross attention fusion block that integrates these per-function representations into a unified global context, propagating information across functions and modelling their continuously evolving, nonlinear couplings, before decoding via a NCDE head to produce the final continuous output.

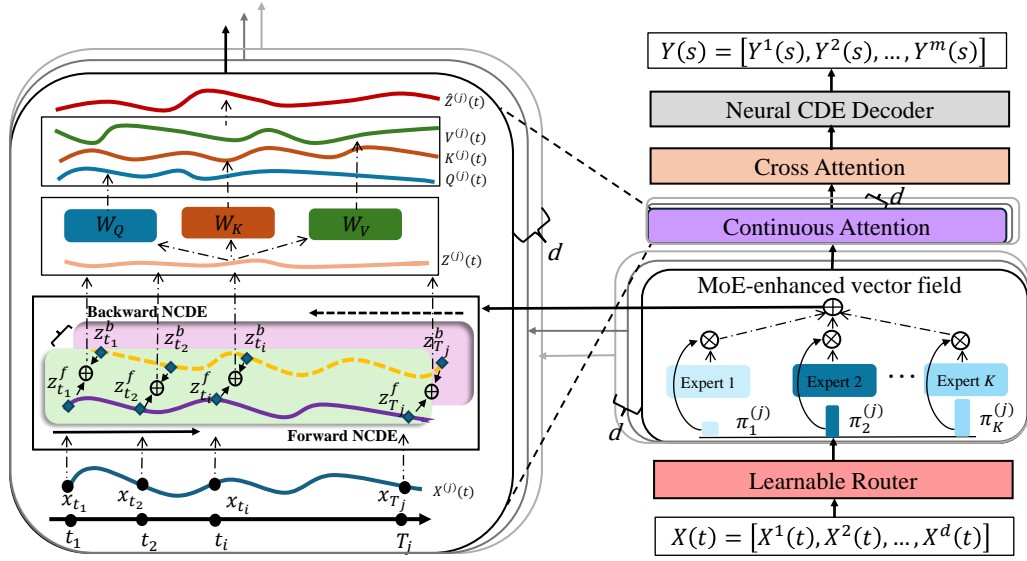

Figure 1: Architecture of the FAME.

## 4.1 Capturing intra-functional continuity through continuous attention

**Controlled differential equations.** The core motivation of our method is to construct attention mechanisms that operate directly on continuous functional data. We adopt the framework of neural controlled differential equations (NCDEs), whose integrals are interpreted in the Young sense because every input function has finite 1-variation. In a controlled differential equation (CDE)[15], an external driver continuously influences the internal dynamics, providing a principled route for learning from irregularly sampled paths. Formally, the latent trajectory is defined as the solution of:

$$z(t) = z(t_0) + \int_{t_0}^{t} f_\theta\big(z(u)\big)\, dX(u), \qquad t \in (t_0, T_j], \tag{1}$$

where the initial state $z(t_0) = \xi_\theta\big(X(t_0), t_0\big)$ is learnable, $z(t)$ encodes an evolving summary of the input path, and $f_\theta$ governs how this summary changes in response to the signal $X$.

**Bidirectional NCDE.** A one-sided CDE only observes the driving signal on $[t_0, t]$, so its latent state lacks access to the future segment $(t, T_j]$. To capture the essential global structure of functional data, we propose a bidirectional NCDE that augments the standard forward integration with a complementary backward pass. For each function $X^{(j)}$, we solve a pair of controlled differential equations:

$$
\begin{aligned}
Z_{\text{fwd}}^{(j)}(t) &= Z^{(j)}(t_0) + \int_{t_0}^{t} f_{\theta_j^{\text{fwd}}}\big(Z_{\text{fwd}}^{(j)}(\tau)\big)\, dX^{(j)}(\tau), \\
Z_{\text{bwd}}^{(j)}(t) &= Z^{(j)}(T_j) - \int_{t}^{T_j} f_{\theta_j^{\text{bwd}}}\big(Z_{\text{bwd}}^{(j)}(\tau)\big)\, d\tilde{X}^{(j)}(\tau),
\end{aligned}
\qquad t \in [t_0, T_j], \tag{2}
$$

where $f_{\theta_j^{\text{fwd}}}$ and $f_{\theta_j^{\text{bwd}}}$ are direction-specific vector fields, and $\tilde{X}^{(j)}$ denotes the reversed input function $\tilde{X}^{(j)}(t) = X^{(j)}(T_j - t)$.

**Assumption 1** (Directional Lipschitz regularity). *Each vector field is globally Lipschitz: that is, $f_{\theta_j^{\text{fwd}}}$ is $L_j^{fwd}$-Lipschitz and $f_{\theta_j^{\text{bwd}}}$ is $L_j^{bwd}$-Lipschitz.*

**Theorem 1** (Existence and uniqueness of the Bi-NCDE). *Under Assumption 1, both the forward and backward CDEs in (2) admit unique solutions on $[t_0, T_j]$. Moreover, letting $L_j = \max\{L_j^{fwd}, L_j^{bwd}\}$, the latent paths satisfy*

$$\big\| Z^{(j)} - \tilde{Z}^{(j)} \big\|_\infty \;\leq\; e^{L_j(T_j - t_0)} \big\| X^{(j)} - \tilde{X}^{(j)} \big\|_{\text{1-var}}.$$

Then, the final latent representation is defined as $Z^{(j)}(t) = [Z_{\text{fwd}}^{(j)}(t), Z_{\text{bwd}}^{(j)}(t)] \in \mathbb{R}^{2h}$, combining local dynamics with global context.

**Continuous attention.** To extend attention from discrete sequences to continuous functions, we apply three bounded matrices $W_Q, W_K, W_V \in \mathbb{R}^{d_f \times 2h}$ to $Z^{(j)}(t)$, thereby obtaining continuous Query, Key, and Value functions:

$$Q^{(j)}(t) = W_Q Z^{(j)}(t), \quad K^{(j)}(t) = W_K Z^{(j)}(t), \quad V^{(j)}(t) = W_V Z^{(j)}(t), \quad t \in [t_0, T_j]. \quad (3)$$

Because bounded linear maps preserve Young-integrability and Lipschitz regularity, the ensuing attention operations are well defined over $[t_0, T_j]$ and reserve the continuous structure of the data. For any $(t, \tau)$, we compute the attention score and normalized weight

$$\alpha^{(j)}(t, \tau) = \frac{\langle Q^{(j)}(t), K^{(j)}(\tau) \rangle}{\sqrt{d_f}}, \qquad \hat{\alpha}^{(j)}(t, \tau) = \frac{e^{\alpha^{(j)}(t, \tau)}}{\displaystyle\int_{t_0}^{T_j} e^{\alpha^{(j)}(t, u)} \, du}, \qquad (4)$$

and form the attended representation

$$\hat{Z}^{(j)}(t) = \int_{t_0}^{T_j} \hat{\alpha}^{(j)}(t, \tau) \, V^{(j)}(\tau) \, d\tau, \qquad t \in [t_0, T_j]. \quad (5)$$

**Assumption 2** (Normalization regularity). *There exists $\gamma > 0$ such that $\int_{t_0}^{T_j} e^{\alpha^{(j)}(t,s)} \, ds \geq \gamma$ for all $t$ and admissible inputs. In practice we ensure this by a temperature parameter $\tau^{\text{temp}} \geq \tau_0^{\text{temp}} > 0$ and log-sum-exp stabilisation.*

**Proposition 1** (Lipschitz stability). *Under Assumptions 1 and 2,*

$$\left\| \hat{Z}^{(j)} - \hat{\tilde{Z}}^{(j)} \right\|_\infty \leq \frac{\|W_V\|}{\gamma} e^{L_j(T_j - t_0)} L_j (T_j - t_0) \left\| X^{(j)} - \tilde{X}^{(j)} \right\|_{1\text{-}var}.$$

*Denote the constant on the right by $L_{\text{single}}$ for later reference.*

**Proposition 2** (Universal approximation). *Let $k \in C([t_0, T_j]^2)$ and $\varepsilon > 0$. If $d_f \geq h$ and the width of $f_\theta$ is sufficiently large, then there exist parameters $(f_\theta, W_Q, W_K, W_V)$ with $\left\| \alpha^{(j)} - k \right\|_\infty < \varepsilon$. Hence $X^{(j)} \mapsto \hat{Z}^{(j)}$ can approximate any bounded continuous operator on functions, independent of the sampling grid;*

Proposition 1 confirms the Lipschitz stability of our continuous attention mechanism: a small perturbation of the input function produces only a proportionally small change in its attended representation. Moreover, Proposition 2 shows that, with sufficient width, the continuous attention can approximate any bounded continuous kernel and thus emulate arbitrary integral operators in FoFR. Consequently, by embedding continuous attention within a bidirectional NCDE, our model not only captures both local dynamics and global context but also can realize any mapping in the function space. More details are provided in Appendix A.

### 4.2 Encoding feature-wise heterogeneity with MoE-Driven Bi-NCDE

Real-world functional data exhibit evident feature heterogeneity, as each function may vary on its own scale, frequency, and noise level. A single vector field can scarcely accommodate such diversity. To address this, we assign each function its own MoE vector field. Specifically, a shared router computes, once per function, a fixed convex combination of $K$ specialist fields, which then governs the bidirectional NCDE dynamics for that function.

**Function-wise router and expert fields.** Let $\{f_{\theta_k} : \mathbb{R}^h \to \mathbb{R}^h\}_{k=1}^K$ be $K$ expert vector fields. For each input function $X^{(j)}$ we compress its functional information with a lightweight 1-D convolution followed by global average pooling, obtaining a summary vector $s^{(j)} = \text{ConvPool}(X^{(j)}) \in \mathbb{R}^{h_0}$. The router $g_\phi : \mathbb{R}^{h_0} \to \mathbb{R}^K$ assigns softmax weights $\pi^{(j)} = \text{softmax}(g_\phi(s^{(j)})) \in \Delta^{K-1}$ to the $K$ specialist fields, yielding the MoE vector field

$$f_\Theta^{(j)}(z) = \sum_{k=1}^K \pi_k^{(j)} f_{\theta_k}(z), \qquad \Theta = \{\theta_1, \ldots, \theta_K, \phi\}. \quad (6)$$

**Directional experts with a shared router.** To preserve the bidirectional representation we instantiate disjoint expert sets $\{f_{\theta_k^{\text{fwd}}}\}$ and $\{f_{\theta_k^{\text{bwd}}}\}$, yet reuse the same gates $\pi^{(j)}$ in both directions:

$$f_{\Theta^{\text{fwd}}}^{(j)}(z) = \sum_k \pi_k^{(j)} f_{\theta_k^{\text{fwd}}}(z), \qquad f_{\Theta^{\text{bwd}}}^{(j)}(z) = \sum_k \pi_k^{(j)} f_{\theta_k^{\text{bwd}}}(z).$$

**Assumption 3** (MoE regularity). *Each expert $f_{\theta_k^{\text{fwd}}}$ and $f_{\theta_k^{\text{bwd}}}$ is globally $L_k^{fwd}$- and $L_k^{bwd}$-Lipschitz, respectively, and bounded by $B_k^{fwd}$ and $B_k^{bwd}$. The router $g_\phi$ is bounded on the summary space: $\|g_\phi(s)\|_\infty \le B_g$ for all $s \in \mathbb{R}^{h_0}$.*

**Lemma 1** (Mixed-field Lipschitz bound). *Under Assumption 3, the mixed fields $f_{\Theta^{\text{fwd}}}^{(j)}$ and $f_{\Theta^{\text{bwd}}}^{(j)}$ are globally Lipschitz with*

$$L_{mix} = \max\Big\{ \sum_k \pi_k^{(j)} L_k^{fwd}, \ \sum_k \pi_k^{(j)} L_k^{bwd} \Big\} \ \le \ \max_k \big( L_k^{fwd}, L_k^{bwd} \big).$$

*Consequently, the bidirectional NCDE driven by (6) satisfies the existence, uniqueness, and stability properties of Theorem 1.*

**MoE-Driven Bi-NCDE.** Replacing $f_{\theta_j^{\text{fwd}}}$ and $f_{\theta_j^{\text{bwd}}}$ in (2) with $f_{\Theta^{\text{fwd}}}^{(j)}$ and $f_{\Theta^{\text{bwd}}}^{(j)}$ yields the MoE-Driven Bi-NCDE, which enables each input function to employ a custom vector field mixture and thereby model input functions with markedly different characteristics.

**Theorem 2** (Well-posed MoE-Driven Bi-NCDE). *The MoE augmentation admits unique solutions, and the stability bound (1) holds with $L_j = \max\{L_{mix}^{fwd}, L_{mix}^{bwd}\}$.*

Thus the MoE design endows the encoder with function-wise expressiveness while retaining all continuity, sampling invariance, and Lipschitz stability guarantees of the base architecture.

### 4.3 Integrating inter-functional interactions via cross attention

Having obtained a function-wise representation $\hat{Z}^{(j)}$ for each input function, we now capture the inter-functional dependencies that single-channel encoders miss. We realise this with a multi-head cross attention that, at every time point $t$, allows each function to attend to the concurrent states of all other functions.

**Cross attention.** For every head $p \in \{1, \ldots, M\}$ and function $j \in \{1, \ldots, d\}$, shared linear maps $W_Q^{(p)}, W_K^{(p)}, W_V^{(p)} \in \mathbb{R}^{d_c \times d_f}$ project the latent paths into query, key, and value functions:

$$Q^{(j,p)}(t) = W_Q^{(p)} \hat{Z}^{(j)}(t), \quad K^{(j,p)}(t) = W_K^{(p)} \hat{Z}^{(j)}(t), \quad V^{(j,p)}(t) = W_V^{(p)} \hat{Z}^{(j)}(t). \tag{7}$$

Cross attention weights are defined by

$$\hat{\beta}^{(j,\ell,p)}(t) = \frac{\exp\{\langle Q^{(j,p)}(t), K^{(\ell,p)}(t)\rangle / \sqrt{d_c}\}}{\displaystyle\sum_{r=1}^{d} \exp\{\langle Q^{(j,p)}(t), K^{(r,p)}(t)\rangle / \sqrt{d_c}\}}, \qquad \ell \in \{1, \ldots, d\}. \tag{8}$$

and each head outputs

$$H^{(j,p)}(t) = \sum_{\ell=1}^{d} \hat{\beta}^{(j,\ell,p)}(t) \, V^{(\ell,p)}(t). \tag{9}$$

Concatenating the $M$ heads and applying $W_O \in \mathbb{R}^{2h \times M d_c}$ gives

$$H^{(j)}(t) = W_O \big[ H^{(j,1)}(t) \| \ldots \| H^{(j,M)}(t) \big], \qquad t \in [t_0, T_j]. \tag{10}$$

Here $W_O$ projects the concatenated representation back to the original $2h$-dimensional latent space, keeping the hidden width uniform throughout the network.

**Assumption 4** (Cross attention Lipschitz). *For every head $p \in \{1, \ldots, M\}$ the projection matrices satisfy $\|W_Q^{(p)}\|_{\text{op}}, \|W_K^{(p)}\|_{\text{op}}, \|W_V^{(p)}\|_{\text{op}}, \|W_O\|_{\text{op}} \le M_{mat}$ for some constant $M_{mat} > 0$. Under this condition, the cross attention map $\hat{Z} \mapsto H$ is globally $L_{cross}$-Lipschitz with $L_{cross} \le (M_{mat})^2 / \sqrt{d_c}$ with respect to the 1-variation norm on function space; see Appendix A.1 for the derivation.*

**CDE decoder for functional targets.** To produce continuous functional outputs, we feed the stacked path $H(t) = \left[ H^{(1)}(t) \| \ldots \| H^{(d)}(t) \right] \in \mathbb{R}^{2hd}$ into a shallow neural CDE:

$$\hat{Y}^{(\zeta)}(s) = y^{(\zeta)}(s_0) + \int_0^s f_\psi\big(\hat{Y}^{(\zeta)}(u)\big)\, dH(u), \qquad s \in [0, S_\zeta], \tag{11}$$

where $f_\psi : \mathbb{R}^m \to \mathbb{R}^m$ is a learnable vector field. Because the solution of (11) can be evaluated at *any* index $s$, the decoder naturally accommodates misaligned or irregular target grids.

**Assumption 5** (Decoder Lipschitz regularity). *The decoder field $f_\psi$ is globally $L_\psi$-Lipschitz and bounded: $\|f_\psi\|_\infty \le B_\psi$.*

**Theorem 3** (Well-Posedness and Lipschitz stability of the decoder). *Under Assumptions 4–5, the encoder output path $H$ has finite $1$-variation, and (11) admits a unique solution $\hat{Y}^{(\zeta)} \in C\big([0, S_\zeta], \mathbb{R}\big)$ with*

$$\|\hat{Y}^{(\zeta)} - \tilde{Y}^{(\zeta)}\|_\infty \le e^{L_\psi S_\zeta} L_\psi S_\zeta \, \|H - \tilde{H}\|_{1\text{-}var}$$

Hence, for each output index $\zeta$, the decoder $\mathcal{D}_\psi : H \mapsto \hat{Y}^{(\zeta)}$ is a bounded, continuous operator from $\big(C([0, S_\zeta], \mathbb{R}^{2hd}), \|\cdot\|_{1\text{-var}}\big)$ to the Banach space $\big(C([0, S_\zeta], \mathbb{R}), \|\cdot\|_\infty\big)$.

**End-to-end Lipschitz bound and generalization.** Composing the encoder $\mathcal{E}_\theta : X \mapsto H$ (continuous and cross attention) with the decoder $\mathcal{D}_\psi : H \mapsto \hat{Y}$ yields the operator $\mathcal{T}_\Theta = \mathcal{D}_\psi \circ \mathcal{E}_\theta$. Denote $S_{\max} = \max_\zeta S_\zeta$. Under the preceding assumptions, $\mathcal{T}_\Theta$ is globally Lipschitz with

$$L_* = L_{\text{enc}}\, e^{L_\psi S_{\max}} L_\psi S_{\max}, \qquad L_{\text{enc}} = L_{\text{cross}}\, L_{\text{single}}.$$

This single constant summarizes the theory behind FAME: the model is provably stable, invariant to sampling grids, and enough to represent any continuous operator on functions. Moreover, the hypothesis class has Rademacher complexity $\mathcal{O}\big(L_*/\sqrt{N}\big)$, supplying a quantitative generalization guarantee for FoFR. Further details and auxiliary lemmas are provided in Appendix A.

# 5 Experiments

To evaluate the proposed model, we benchmark FAME against a wide range of state-of-the-art methods for FoFR (see Section 2). For basis–expansion pipelines [38], we pair two orthonormal systems—B-splines and Fourier functions—with four classical regressors: Ordinary Linear, Ridge, Lasso, and Elastic Net. Beyond these eight variants, we include functional principal component analysis (FPCA) [8], operator-valued kernel regression [14], Gaussian processes [32], and the functional neural network (FNN) [36] as stronger baselines. We evaluate FAME and the baselines on both synthetic datasets and several real-world datasets, using mean-squared error (MSE) as the primary evaluation metric.

## 5.1 Datasets

**Synthetic data.** We generate functional data using Gaussian-process trajectories because GP paths have finite $1$-variation almost surely, hence $X_i \in \mathcal{X}$. Specifically, we draw $X_i^{(j)}(t) \sim \mathcal{GP}\big(0, K(t, t')\big)$ for $j = 1, \ldots, d$ and $t \in [0, 1]$, and define the response by $Y_i(t) = \sin\big(\sum_{j=1}^d X_i^{(j)}(t)\big) + \lambda\, \varepsilon_i(t)$, with $\varepsilon_i(t) \overset{\text{i.i.d.}}{\sim} \mathcal{N}(0, \sigma^2)$ and noise level $\lambda \ge 0$. Input and output are sampled on (possibly distinct) irregular grids $\{t_{i,\ell}\}_{\ell=1}^{\Lambda_i}$ and $\{s_{i,r}\}_{r=1}^{\Gamma_i}$. We set $d = 3$, $m = 1$, $\Lambda_i = \Gamma_i = 20$ and $N_s = 200$ unless otherwise specified. In particular, we evaluate FAME under seven controlled settings. **Case 1** fixes the input and output grids at $\Lambda_i = \Gamma_i = 20$, perfectly aligned across all samples. **Case 2** increases the shared resolution to $\Lambda_i = \Gamma_i = 50$, while preserving grid alignment. **Case 3** keeps the output grid at $\Gamma_i = 20$ but samples the input resolution independently as $\Lambda_i \in \{10, 20, 50\}$ for each sample, thereby introducing variable input granularity. Throughout Cases 1–3 we vary the dataset size $N_s \in \{100, 200, 500\}$ to disentangle the effects of resolution and sample size. Moving to stress tests, **Case 4** introduces feature heterogeneity by assigning each input function a distinct RBF kernel width $\sigma_j \in \{0.2, 0.3, 0.5\}$, rather than the fixed value of $0.3$ used in the other cases. **Case 5** probes noise robustness by adding independent Gaussian perturbations of noise level $\lambda \in \{0.1, 0.2, 0.3\}$ to every output. **Case 6** tests multivariate prediction by extending the target

Table 1: Average test MSE for different methods in regression. Detailed results (mean $\pm$ standard deviation) are provided in Appendix B. The best MSE for each case is highlighted in bold.

| Model | | Case 1 | | | Case 2 | | | Case 3 | | |
|---|---|---|---|---|---|---|---|---|---|---|
| | | 100 | 200 | 500 | 100 | 200 | 500 | 100 | 200 | 500 |
| B-spline | Linear | 0.4720 | 0.3947 | 0.3123 | 0.4135 | 0.3412 | 0.2822 | 0.4958 | 0.4002 | 0.3260 |
| | Ridge | 0.4264 | 0.3893 | 0.3117 | 0.3960 | 0.3393 | 0.2817 | 0.3869 | 0.3740 | 0.3222 |
| | Lasso | 0.4098 | 0.3830 | 0.3052 | 0.3856 | 0.3351 | 0.2766 | 0.4132 | 0.3584 | 0.3188 |
| | Elastic Net | 0.4510 | 0.3650 | 0.2874 | 0.3725 | 0.3152 | 0.2583 | 0.3986 | 0.3618 | 0.3190 |
| Fourier | Linear | 0.5002 | 0.4092 | 0.3224 | 0.4923 | 0.3636 | 0.2896 | 0.4762 | 0.3921 | 0.3547 |
| | Ridge | 0.4255 | 0.3780 | 0.3149 | 0.3616 | 0.3343 | 0.2841 | 0.3755 | 0.3568 | 0.3328 |
| | Lasso | 0.3550 | 0.3493 | 0.3135 | 0.3361 | 0.3280 | 0.3001 | 0.3808 | 0.3560 | 0.3247 |
| | Elastic Net | 0.3540 | 0.3325 | 0.2914 | 0.3167 | 0.3070 | 0.2720 | 0.3737 | 0.3496 | 0.3366 |
| FPCA | | 0.3717 | 0.3554 | 0.3200 | 0.2890 | 0.2733 | 0.2624 | 0.3812 | 0.3563 | 0.3295 |
| Kernel Method | | 0.2441 | 0.1728 | 0.1058 | 0.1741 | 0.0923 | 0.0700 | 0.2654 | 0.2445 | 0.1485 |
| Gaussian Process | | 0.3405 | 0.2941 | 0.2036 | 0.3905 | 0.3917 | 0.2588 | 0.3031 | 0.2926 | 0.3945 |
| FNN | | 0.3123 | 0.2083 | 0.1013 | 0.1941 | 0.1142 | 0.0811 | 0.3678 | 0.3571 | 0.1366 |
| FAME w/o Bi-dir | | 0.1832 | 0.0812 | 0.0654 | 0.1530 | 0.0528 | 0.0355 | 0.1919 | 0.0813 | 0.0368 |
| FAME w/o MoE | | 0.1870 | 0.0828 | 0.0663 | 0.1578 | 0.0538 | 0.0362 | 0.1972 | 0.0856 | 0.0374 |
| FAME w/o Cross-attn | | 0.1902 | 0.0815 | 0.0668 | 0.1602 | 0.0544 | 0.0375 | 0.1997 | 0.0879 | 0.0381 |
| **FAME** | | **0.1806** | **0.0783** | **0.0635** | **0.1532** | **0.0511** | **0.0342** | **0.1954** | **0.0796** | **0.0352** |

Table 2: Average test set MSE in regression under simulation. Basis Expansion (best) shows the best result among the 8 basis expansion methods presented in Table 1.

| Model | case 4 | case 5 | | | case 6 | case 7 | | case 8 |
|---|---|---|---|---|---|---|---|---|
| | | 0.1 | 0.2 | 0.3 | | 5 | 10 | |
| Basis Expansion(best) | 0.3610 | 0.3665 | 0.3944 | 0.4419 | 0.3669 | 0.4501 | 0.4705 | 0.5204 |
| FPCA | 0.3802 | 0.3879 | 0.3956 | 0.4570 | 0.3844 | 0.5284 | 0.5573 | 0.5737 |
| Kernel Method | 0.1928 | 0.1440 | 0.1919 | 0.2435 | 0.2022 | 0.4659 | 0.5236 | 0.5895 |
| Gaussian Process | 0.2302 | 0.3079 | 0.3356 | 0.3830 | 0.3434 | 0.4120 | 0.4498 | 0.4762 |
| FNN | 0.2102 | 0.1879 | 0.2250 | 0.3438 | 0.1744 | 0.4801 | 0.5164 | 0.5384 |
| FAME | **0.0798** | **0.0846** | **0.1076** | **0.1420** | **0.0824** | **0.2285** | **0.3330** | **0.3530** |

to the two-dimensional map $[\sin(\sum_j X_i^{(j)}(t)), \ \cos(\sum_j X_i^{(j)}(t))]^\top$. **Case 7** stresses scalability by increasing the number of input functions to $d \in \{5, 10\}$ while keeping all other settings unchanged. Finally, to isolate the impact of the hyperparameter $K$ under high dimensionality and heterogeneity, **Case 8** fixes the number of input functions at $d = 10$, generates inputs with RBF kernels of widths $\sigma_j \in \{0.2, 0.3, 0.5\}$, and varies $K \in \{1, 2, 3, 5, 8\}$.

**Real-world data.** We evaluate the same set of models on three public datasets: 1) *Hawaii Ocean*, which contains five hydrographic depth profiles—temperature, salinity, oxygen, chloropigment, and density—among which different variables are treated as regression targets in turn, with the remaining serving as input functions; 2) *Human3.6M*, a human motion capture dataset consisting of 3-D joint trajectories, where we define three action-specific regression tasks (Walking, Eating, and Sitting); and 3) *ETT-small*, a monthly electricity-transformer dataset used to forecast oil temperature from transformer load curves. Full preprocessing details and task definitions are provided in Appendix B.

## 5.2 Experimental Results

Table 1 summarizes the performance of FAME and various baselines on the synthetic datasets, clearly demonstrating that FAME consistently achieves the lowest test-set MSE. Other methods based on fixed basis expansions, FPCA, Gaussian processes, or neural networks achieve inferior performance,

Table 3: Average test set MSE in regression on different real-world datasets.

| Model | Hawaii ocean | | Human3.6M | | | ETDataset |
| | Salinity | Temp | Walking | Eating | Sitting down | Oil Temp |
| --- | --- | --- | --- | --- | --- | --- |
| Basis Expansion(best) | 0.0780 | 0.0014 | 0.0359 | 0.04841 | 0.0122 | 0.0365 |
| FPCA | 0.0865 | 0.0025 | 0.0373 | 0.0099 | 0.0121 | 0.0355 |
| Kernel Method | 0.0754 | 0.0025 | 0.0373 | 0.0099 | 0.0121 | 0.0355 |
| Gaussian Process | 0.0931 | 0.0022 | 0.0360 | 0.0075 | 0.0107 | 0.0380 |
| FNN | 0.0766 | 0.0020 | 0.0373 | 0.0099 | 0.0121 | 0.0355 |
| FAME w/o Bi-dir | 0.0751 | 0.0012 | 0.0327 | 0.0035 | 0.0071 | 0.0264 |
| FAME w/o MoE | 0.0759 | 0.0013 | 0.0332 | 0.0038 | 0.0075 | 0.0271 |
| FAME w/o Cross-attn | 0.0773 | 0.0014 | 0.0344 | 0.0044 | 0.0083 | 0.0286 |
| FAME | **0.0748** | **0.0012** | **0.0325** | **0.0034** | **0.0070** | **0.0262** |

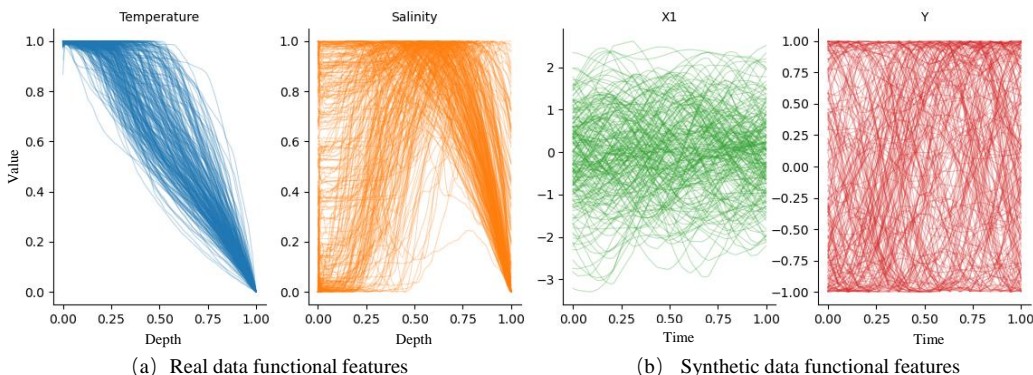

(a) Real data functional features      (b) Synthetic data functional features

Figure 2: Visualization of Functional Features.

largely due to their limited capacity to adapt to nonlinear functional mappings, sensitivity to irregular sampling points, or challenges in preserving continuous functional structures. In contrast, FAME's continuous attention mechanism, supported by adaptive MoE-driven dynamics and inter-dimensional cross attention, provides a stable and expressive mapping that directly operates in continuous function spaces, which the ablation study in Table 1 clearly demonstrates. Table 2 further evaluates model robustness across a variety of stress-test scenarios—including feature heterogeneity, noise levels, output structures, and input dimensionalities—demonstrating that FAME consistently maintains superior predictive performance under challenging conditions. Table 3 reports results on real-world datasets, confirming the practical applicability of our model. The performance advantage over strong baselines is reduced compared to the synthetic setting. Figure 2(a) shows that the HAWAII OCEAN dataset occupies a smooth, low-variability subspace dictated by ocean stratification, a structure that fixed-basis and kernel methods can already approximate competently. Figure 2(b) contrasts this with synthetic trajectories drawn from independent Gaussian processes, whose broad, irregular function space contains pronounced local fluctuations. Under this more demanding regime, the continuous-attention encoder in FAME secures a substantially larger accuracy gain, underscoring its flexibility across disparate function-space complexities and corroborating the universal-approximation and sampling-invariance guarantees established in Section 4.3.

As shown in Figure 3, the model's performance varies systematically with key parameters, illustrating clear relationships between predictive behavior and data characteristics. Specifically, in Case 1, the observed test error progressively decreases with increasing sample size (Fig. 3(a)), in agreement with our theoretical generalisation result. Case 2 illustrates a clear improvement in model performance as the number of sampling points grows, with FAME achieving superior accuracy even at a sparse resolution of 10 sampling points (Fig. 3(b)). Case 3 reveals that while baseline methods experience substantial performance degradation under mixed input resolutions, FAME maintains stable and

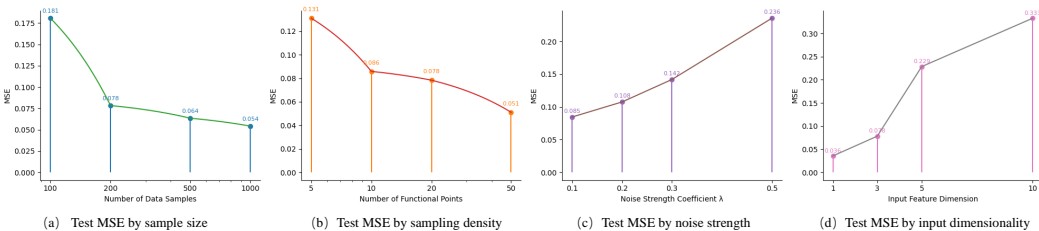

(a) Test MSE by sample size     (b) Test MSE by sampling density     (c) Test MSE by noise strength     (d) Test MSE by input dimensionality

Figure 3: Parameter-sensitivity curves for FAME.

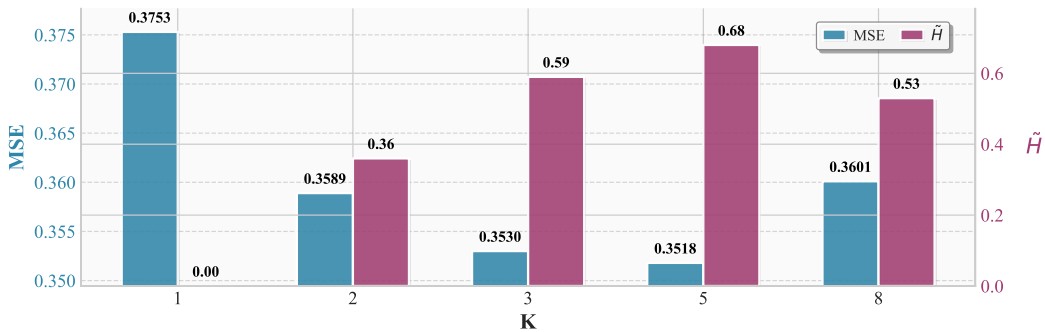

Figure 4: Sensitivity to $K$—test MSE (left axis) and $\tilde{H}$ (right axis).

superior performance. This result highlights our model's sampling invariance capability, endowed by the Young integral formulation. As the input becomes noisier or more high-dimensional (Cases 4 and 7), all models exhibit increased error (Fig. 3(c),(d)), but FAME remains within a practically useful error range. Taken together, these parameter sweeps confirm that FAME's empirical behaviour matches its theoretical guarantees, maintaining robust accuracy across diverse data regimes. To assess how the number of experts $K$ affects performance, we plot the test MSE together with the normalized routing entropy $\tilde{H} = H/\log K$ in Figure 4, where $H$ denotes the Shannon entropy of the gating distribution and the normalization by $\log K$ yields a $[0, 1]$-scaled quantity comparable across different $K$. The curves show that accuracy improves as $K$ increases from 1 to 3 and plateaus around $K = 3 \sim 5$, while $\tilde{H}$ remains moderate—indicating healthy, non-collapsed expert utilization. Beyond that, accuracy does not meaningfully improve and computation grows with $K$. Consequently, $K = 3$ and $K = 5$ show stable accuracy and healthy expert utilization; since computation scales with $K$, we adopt $K = 3$ as the default. More extensive visualizations and discussions are provided in Appendix B, which also details the experimental setup ( hyperparameters, and training schedules) and a comprehensive analysis of computational efficiency (time and memory complexity).

## 6 Conclusion

We have presented **FAME**, a fully end-to-end framework that learns FoFR mappings by coupling bidirectional neural CDEs with a continuous attention mechanism, enriching the resulting latent dynamics through a mixture-of-experts router, and fusing inter-functional information via multi-head cross attention before decoding with a NCDE. This design yields a resolution-agnostic, Lipschitz-stable, and universally expressive operator that consistently outperforms classical bases, RKHS models, and recent deep networks on both synthetic and real benchmarks. A natural limitation of FAME is that when the underlying operator varies little across the function domain, simpler linear or low-rank models may offer a more attractive trade-off between statistical and computational complexity. In future work, we plan to extend the same functional attention principle beyond FoFR to tasks such as functional classification and function-on-scalar prediction, where the continuity-aware architecture of FAME is expected to provide similar advantages.

## Acknowledgement

Yifei Gao and Chen Zhang would like to acknowledge the support of NSFC Grant 72271138 and Tsinghua–NUS Joint Funding 20243080039.

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

# A Theoretical Guarantees of FAME

This appendix presents complete proofs for all theoretical claims made in the main text. We begin by deriving Lipschitz constants for each architectural block and compose them to obtain an end-to-end stability bound. We then establish sampling invariance and universal approximation, and conclude with a quantitative generalisation result based on Rademacher complexity.

**Notation.** Each input function $X^{(j)}$ is defined on its own horizon $[0, T_j]$; we write

$$\mathcal{X} \;=\; \prod_{j=1}^{d} C([0, T_j], \mathbb{R}), \qquad \|X\|_{1\text{-var}} \;=\; \max_{j} \|X^{(j)}\|_{1\text{-var}}.$$

Each output function $Y^{(\zeta)}$ lives on $[0, S_\zeta]$; correspondingly,

$$\mathcal{Y} \;=\; \prod_{\zeta=1}^{m} C([0, S_\zeta], \mathbb{R}), \qquad \|Y\|_\infty \;=\; \max_{\zeta} \|Y^{(\zeta)}\|_\infty.$$

For any (vector-valued) path $X$, $\|X\|_{1\text{-var}}$ denotes the usual total variation and is taken component-wise as above. All integrals are understood in the Young sense [46], so they depend only on the underlying continuous functions, not on the choice of discretisation.

## A.1 Module-wise Lipschitz constants

**(i) Continuous attention.** Under Assumption 1 the bidirectional NCDE (forward *and* backward flows) is well posed for every input function. Theorem 1 in the main text shows that the mapping $X \mapsto \hat{Z}$ satisfies

$$\|\hat{Z} - \tilde{\hat{Z}}\|_\infty \;\le\; L_{\text{single}} \|X - \tilde{X}\|_{1\text{-var}}, \qquad L_{\text{single}} = \frac{\|W_V\|}{\gamma} e^{L_j(T_j - t_0)} L_j(T_j - t_0).$$

**(ii) Cross attention.** Let $\sigma(a) = \text{softmax}\big(a/\sqrt{d_c}\big)$.

**Lemma 2** (Soft-max contraction). *The map $\sigma : \mathbb{R}^{d_c} \to \Delta^{d_c - 1}$ is $1/\sqrt{d_c}$-Lipschitz from $(\ell_\infty, \|\cdot\|_\infty)$ to $(\ell_1, \|\cdot\|_1)$.*

*Proof.* The Jacobian of $\sigma$ is $\text{diag}(\sigma) - \sigma\sigma^\top$; its operator norm $\ell_\infty \to \ell_1$ equals $1/\sqrt{d_c}$. Integrating this bound along the line segment joining any two inputs yields the claim. $\qquad\square$

If the projection matrices satisfy $\|W_Q^{(p)}\|_{\text{op}}, \|W_K^{(p)}\|_{\text{op}}, \|W_V^{(p)}\|_{\text{op}}, \|W_O\|_{\text{op}} \le M_{mat}$ for every head $p$, Lemma 2 implies, path-wise in 1-variation,

$$\|H - \tilde{H}\|_{1\text{-var}} \;\le\; L_{\text{cross}} \|\hat{Z} - \tilde{\hat{Z}}\|_{1\text{-var}}, \qquad L_{\text{cross}} = \frac{M_{mat}^3 M}{\sqrt{d_c}}.$$

**(iii) CDE decoder.** If $f_\psi$ is globally $L_\psi$-Lipschitz and bounded, the Young–Löwner framework together with Grönwall's inequality yields, for every output function $\zeta$,

$$\|\hat{Y}^{(\zeta)} - \tilde{Y}^{(\zeta)}\|_\infty \;\le\; L_{\text{dec}} \|H - \tilde{H}\|_{1\text{-var}}, \qquad L_{\text{dec}} = e^{L_\psi S_\zeta} L_\psi S_\zeta.$$

Writing $S_{\max} = \max_\zeta S_\zeta$, the end-to-end Lipschitz radius is

$$L_* = L_{\text{dec}} L_{\text{cross}} L_{\text{single}}, \qquad L_{\text{dec}} = e^{L_\psi S_{\max}} L_\psi S_{\max}.$$

## A.2 End-to-end stability and sampling invariance

**Theorem 4** (Global Lipschitz bound). *For the composite operator $\mathcal{T}_\Theta = \mathcal{D}_\psi \circ \mathcal{E}_\theta$,*

$$\|\mathcal{T}_\Theta(X) - \mathcal{T}_\Theta(\tilde{X})\|_\infty \;\le\; L_* \|X - \tilde{X}\|_{1\text{-var}}.$$

*Proof.* Chain the inequalities of Section A.1 along $X \to \hat{Z} \to H \to \hat{Y}$. □

**Proposition 3** (Sampling invariance). *If two observation grids encode the same underlying functions, the operator $\mathcal{T}_\Theta$ returns identical outputs.*

*Proof.* Young integrals depend only on the driving functions, not on the chosen partitions [46]. □

### A.3 Universal approximation

**Lemma 3** (Density of separable kernels). *Define*
$$\mathcal{S} = \left\{ k(t, \tau) = \sum_{i=1}^{r} u_i(t)\, v_i(\tau) \;\middle|\; r \in \mathbb{N},\; u_i, v_i \in C([t_0, T_j]) \right\}. \tag{12}$$
*The uniform closure of $\mathcal{S}$ equals the full kernel space:*
$$\overline{\mathcal{S}} = C([t_0, T_j]^2).$$

*Proof. (i) Sub-algebra.* $\mathcal{S}$ is closed under point-wise addition, scalar multiplication, and multiplication, so it is a sub-algebra of $C([t_0, T_j]^2)$.

*(ii) Constants.* Choosing $u \equiv 1$ and $v \equiv c$ gives the constant kernel $k(t, \tau) \equiv c \in \mathcal{S}$.

*(iii) Point separation.* For two distinct points $(t_1, \tau_1) \neq (t_2, \tau_2)$:

- If $t_1 \neq t_2$, pick $u \in C([t_0, T_j])$ with $u(t_1) \neq u(t_2)$ (by Urysohn's lemma) and set $v \equiv 1$.
- If $t_1 = t_2$ (hence $\tau_1 \neq \tau_2$), pick $v \in C([t_0, T_j])$ with $v(\tau_1) \neq v(\tau_2)$ and set $u \equiv 1$.

In both cases the resulting kernel belongs to $\mathcal{S}$ and takes different values at the two points, so $\mathcal{S}$ separates points of the compact Hausdorff space $[t_0, T_j]^2$.

*(iv) Stone–Weierstrass.* Because $\mathcal{S}$ is a sub-algebra that contains the constants and separates points, the real Stone–Weierstrass theorem [34] yields $\overline{\mathcal{S}} = C([t_0, T_j]^2)$. □

**Theorem 5** (Density of the FAME hypothesis class). *For any bounded continuous operator $\mathcal{F} : \mathcal{X} \to \mathcal{Y}$ and any $\varepsilon > 0$ there exists a parameter set $\Theta$ such that*
$$\|\mathcal{T}_\Theta - \mathcal{F}\|_\infty < \varepsilon.$$

*Proof.* Neural CDEs are universal approximators on path space [15]. Lemma 3 shows that cross-attention can approximate any continuous kernel, and the decoder NCDE is a universal curve generator. Universality is preserved under composition, so the class $\{\mathcal{T}_\Theta\}$ is dense in $C(\mathcal{X}, \mathcal{Y})$. □

### A.4 Generalisation bound

**Theorem 6** (Rademacher complexity). *For $N$ i.i.d. samples, the hypothesis class with radius $L_*$ satisfies $\mathfrak{R}_N \leq c\, L_*/\sqrt{N}$ for a universal constant $c$.*

*Proof.* Apply the Ledoux–Talagrand contraction principle [18] to Theorem 4. □

## B Experimental details

### B.1 Real-world Dataset

**Hawaii ocean dataset.** The Hawaii ocean dataset, part of the Hawaii Ocean Time-Series program, includes measurements of hydrographic, chemical, and biological characteristics at a station north of Oahu, Hawaii, from January 1, 1999, to December 31, 2018. The dataset consists of five functional variables: Temperature, Oxygen concentration, Potential density, Salinity, and Chloropigment, measured every 2 meters between 0 and 200 meters below the sea surface. After data preprocessing, we retained 265 samples, with irregular sampling ($\Lambda_i = \Gamma_i = 20$) as the number of observation points for each function sample. For our analysis, Temperature and Salinity alternately serve as the target variable, with the remaining variables used as input predictors.

**Human3.6M dataset.** The Human3.6M dataset [10] consists of 11 subjects performing 15 different actions, with 3D joint coordinates provided for 32 body parts. Each action is captured with high precision across different subjects, making the dataset suitable for studying human motion. In our approach, we enhance the dataset by performing data augmentation through random sampling of time series data from different subjects for the same action. Specifically, for each action, we sample 30 points, and each subject contributes 30 samples, ensuring a diverse set of data for training.

We have designed three specific regression tasks:

- **Walking**: For the *Walking* action, the input consists of the X and Y coordinates of the *right knee*, and the output is the corresponding Z coordinate. This task includes 180 samples from multiple subjects.

- **Eating 1**: For the *Eating 1* action, the input includes the XYZ coordinates of the *right forearm* and the XY coordinates of the *right wrist*, with the output being the Z coordinate of the *right wrist*. This task also contains 180 samples.

- **Sitting down**: For the *Sitting down* and *Sitting down1* actions, the input consists of the XYZ coordinates of *Spine1* and *Spine2*, and the output is the XYZ coordinates of *Spine3*. This task includes 390 samples.

**ETT-small dataset.** The ETT-small dataset [50] contains data from two electricity transformer stations, including variables such as oil temperature and power load. Each sample is constructed from monthly data, comprising 30 data points, with each point representing a day's worth of measurements. For the experiments, we use a total of 48 samples, aiming to assess the performance of regular time sampling in this context.

## B.2 Experimental Setup

All experiments were performed on a workstation equipped with an AMD Ryzen 9 5950X CPU and an NVIDIA RTX 3090 GPU. Unless otherwise stated, every model is trained for 100 epochs with the Adam optimiser, an initial learning rate of $1 \times 10^{-3}$, and a dropout rate of 0.2. Each dataset is randomly split into 80% training and 20% test instances; for the synthetic benchmark we repeat this split five times and report the average performance. For FAME, the neural controlled differential equation that constructs the continuous attention and the decoder CDE share an identical architecture. The input lifting network $\xi_\theta$ and the neural vector field $f_\theta$ are both implemented as two–layer MLPs with hidden widths 32 and 64, respectively, followed by Tanh activations. The same MLP configuration is used for the MoE expert fields and for the decoder vector field $f_\psi$, ensuring architectural consistency throughout the model.

## B.3 Results and discussion

While the main text already provides extensive quantitative evidence, additional results—spanning Tables 4–6 and Figures 5–7—together with analyses of stress tests and computational efficiency, merit inclusion for completeness.

**Supplementary Stress-Test Analysis** Across all supplementary simulations, FAME preserves a decisive lead over competing approaches. The heterogeneity configuration in Case 4 assigns distinct length-scales $\{0.2, 0.3, 0.5\}$ to the three input coordinates; ablating the mixture-of-experts router in this setting raises the test MSE from 0.0798 to 0.0842, demonstrating the value of specialist vector fields for channel-specific dynamics. As the input dimensionality rises (case 7), the test errors of all baselines increase markedly, highlighting how swiftly FoFR becomes more demanding in higher dimensions. Kernel methods, which performed strongly in the low-dimensional settings of Cases 1–3, deteriorate the most. Basis-expansion pipelines and FPCA regressors also degrade, since fixed bases struggle to model the richer cross-coordinate interactions that emerge with additional functional inputs. Although FAME is likewise affected by the added complexity, its relative performance drop is considerably smaller, and its absolute accuracy remains within a practically useful range. These findings indicate that the continuous attention encoder and mixture-of-experts vector fields endow FAME with a robustness that scales more gracefully than existing alternatives when moving to high-dimensional functional spaces.

Table 4: Runtime comparison (training / inference).

| Case 1 (Train / Infer s) | | | |
|---|---|---|---|
| Method | 100 | 200 | 500 |
| Basis Expansion | 0.01 / 0.01 | 0.01 / 0.01 | 0.01 / 0.01 |
| FPCA | 0.01 / 0.01 | 0.01 / 0.01 | 0.01 / 0.01 |
| Kernel Method | 0.01 / 0.01 | 0.01 / 0.01 | 0.01 / 0.01 |
| Gaussian Process | 0.03 / 0.03 | 0.03 / 0.03 | 0.03 / 0.03 |
| FNN | $0.40 \times 10$ / 0.36 | $0.62 \times 10$ / 0.37 | $0.97 \times 10$ / 0.36 |
| FAME | $48.79 \times 20$ (epoch) / 1.02 | $57.46 \times 20$ / 1.00 | $79.53 \times 20$ / 1.03 |
| Case 2 (Train / Infer s) | | | |
| Method | 100 | 200 | 500 |
| Basis Expansion | 0.01 / 0.01 | 0.01 / 0.01 | 0.01 / 0.01 |
| FPCA | 0.01 / 0.01 | 0.01 / 0.01 | 0.01 / 0.01 |
| Kernel Method | 0.01 / 0.01 | 0.01 / 0.01 | 0.01 / 0.01 |
| Gaussian Process | 0.03 / 0.03 | 0.03 / 0.03 | 0.03 / 0.03 |
| FNN | $1.02 \times 10$ / 0.81 | $1.98 \times 10$ / 0.82 | $3.32 \times 10$ / 0.84 |
| FAME | $134.42 \times 20$ / 2.60 | $151.47 \times 20$ / 2.66 | $207.20 \times 20$ / 2.92 |

Table 5: Peak memory usage (MB).

| Method | Basis Expansion | FPCA | Kernel Method | Gaussian Process | FNN | FAME |
|---|---|---|---|---|---|---|
| Memory (MB) | 0.30 | 0.25 | 2.91 | 12.70 | 3.82 | 15.36 |

**Loss Dynamics**   Figure 5 traces the optimisation trajectory of FAME. The loss decreases monotonically and stabilises after roughly 20 epochs, demonstrating both rapid convergence and training stability for the continuous attention architecture.

**Runtime and memory.**   Cases 1–2 jointly span the two principal axes of computational load—the number of input functions and the number of sampling points per function—so they are well suited for reporting cost. All timings in Table 4 were obtained on the same ordinary desktop. In brief, basis/FPCA/kernel methods train fastest when grids are fixed and models are small, but their reliance on pre-specified bases or kernels can limit accuracy. By contrast, FAME is fully data-driven: training is costlier but remains lightweight in absolute terms, and inference latencies are comparable to classical baselines. Even in the most demanding configuration we considered (Case 2 with 500 samples), FAME trains in about 207.20 s per epoch; over 20 epochs this totals $\approx 4144$ s ($\approx 69$ min) on this CPU, while post-training inference remains at the seconds level (Table 4). Memory usage varies little across tasks and is modest overall (Table 5); the peak footprint for FAME is only $\sim 15$ MB, and datasets are purely numeric and small on disk, making deployment straightforward.

**Regression Visualisation**   Figure 6 reports an analogous comparison for the synthetic Case 1 setting. Across the entire domain the predicted and true curves are almost indistinguishable, illustrating FAME's ability to recover fine-grained function-to-function mappings even under irregular sampling. Figure 7 presents model outputs on the *Sitting down* sequence from the Human3.6M benchmark—a representative real-world task. Basis-function baselines capture global trends, whereas conventional neural networks reproduce local variations; FAME accurately follows both scales and achieves the closest alignment with ground truth.

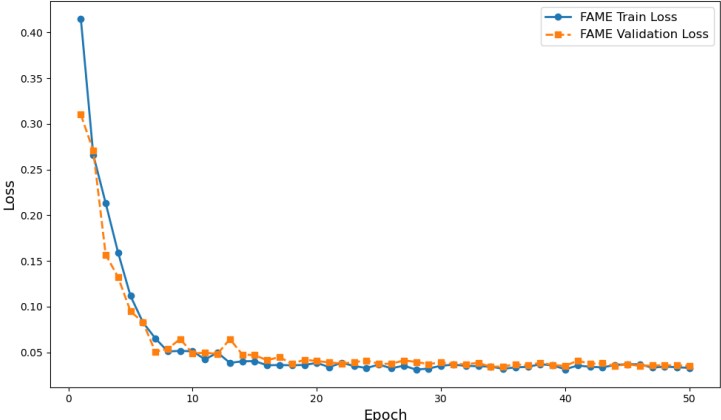

Figure 5: Training and validation loss over epochs. The monotonic convergence illustrates stable optimisation behaviour.

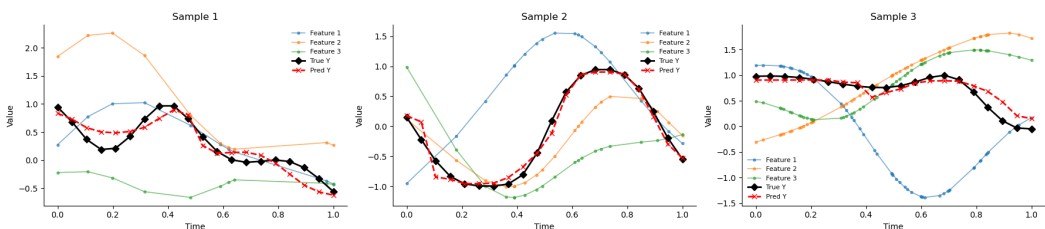

Figure 6: Prediction curves for *Case 1*.

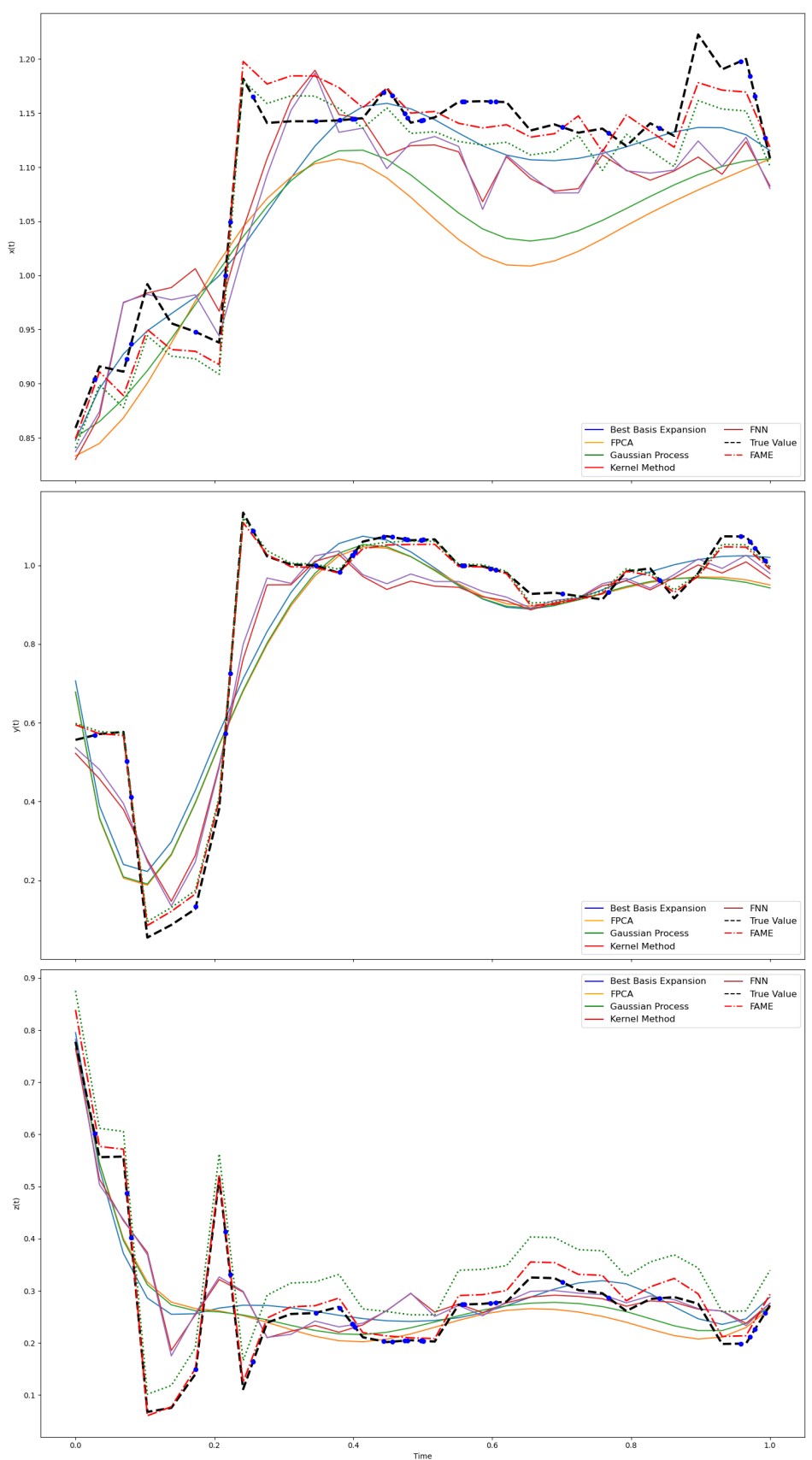

Figure 7: Prediction curves on the *Sitting down* task (Human3.6M).

Table 6: Average test MSE for different methods in regression, reported as mean ± standard deviation. The best MSE for each case is highlighted in bold.

| Model | | Case 1 | | | Case 2 | | | Case 3 | | |
|---|---|---|---|---|---|---|---|---|---|---|
| | | 100 | 200 | 500 | 100 | 200 | 500 | 100 | 200 | 500 |
| B-spline | Linear | 0.4720 ± 0.0378 | 0.3947 ± 0.0316 | 0.3123 ± 0.0250 | 0.4135 ± 0.0331 | 0.3412 ± 0.0273 | 0.2822 ± 0.0226 | 0.4958 ± 0.0397 | 0.4002 ± 0.0320 | 0.3260 ± 0.0261 |
| | Ridge | 0.4264 ± 0.0341 | 0.3893 ± 0.0311 | 0.3117 ± 0.0249 | 0.3960 ± 0.0317 | 0.3393 ± 0.0271 | 0.2817 ± 0.0225 | 0.3869 ± 0.0310 | 0.3740 ± 0.0299 | 0.3222 ± 0.0258 |
| | Lasso | 0.4098 ± 0.0328 | 0.3830 ± 0.0306 | 0.3052 ± 0.0244 | 0.3856 ± 0.0308 | 0.3351 ± 0.0268 | 0.2766 ± 0.0221 | 0.4132 ± 0.0331 | 0.3584 ± 0.0287 | 0.3188 ± 0.0255 |
| | Elastic Net | 0.4510 ± 0.0361 | 0.3650 ± 0.0292 | 0.2874 ± 0.0230 | 0.3725 ± 0.0298 | 0.3152 ± 0.0252 | 0.2583 ± 0.0207 | 0.3986 ± 0.0319 | 0.3618 ± 0.0289 | 0.3190 ± 0.0255 |
| Fourier | Linear | 0.5002 ± 0.0400 | 0.4092 ± 0.0327 | 0.3224 ± 0.0258 | 0.4923 ± 0.0394 | 0.3636 ± 0.0291 | 0.2896 ± 0.0232 | 0.4762 ± 0.0381 | 0.3921 ± 0.0314 | 0.3547 ± 0.0284 |
| | Ridge | 0.4255 ± 0.0340 | 0.3780 ± 0.0302 | 0.3149 ± 0.0252 | 0.3616 ± 0.0289 | 0.3343 ± 0.0267 | 0.2841 ± 0.0227 | 0.3755 ± 0.0300 | 0.3568 ± 0.0285 | 0.3328 ± 0.0266 |
| | Lasso | 0.3550 ± 0.0284 | 0.3493 ± 0.0279 | 0.3135 ± 0.0251 | 0.3361 ± 0.0269 | 0.3280 ± 0.0262 | 0.3001 ± 0.0240 | 0.3808 ± 0.0305 | 0.3560 ± 0.0285 | 0.3247 ± 0.0260 |
| | Elastic Net | 0.3540 ± 0.0283 | 0.3325 ± 0.0266 | 0.2914 ± 0.0233 | 0.3167 ± 0.0253 | 0.3070 ± 0.0246 | 0.2720 ± 0.0218 | 0.3737 ± 0.0299 | 0.3496 ± 0.0280 | 0.3366 ± 0.0269 |
| FPCA | | 0.3717 ± 0.0297 | 0.3554 ± 0.0284 | 0.3200 ± 0.0256 | 0.2890 ± 0.0231 | 0.2733 ± 0.0219 | 0.2624 ± 0.0210 | 0.3812 ± 0.0305 | 0.3563 ± 0.0285 | 0.3295 ± 0.0264 |
| Kernel Method | | 0.2441 ± 0.0195 | 0.1728 ± 0.0138 | 0.1058 ± 0.0085 | 0.1741 ± 0.0139 | 0.0923 ± 0.0074 | 0.0700 ± 0.0056 | 0.2654 ± 0.0212 | 0.2445 ± 0.0196 | 0.1485 ± 0.0119 |
| Gaussian Process | | 0.3405 ± 0.0272 | 0.2941 ± 0.0235 | 0.2036 ± 0.0163 | 0.3905 ± 0.0312 | 0.3917 ± 0.0313 | 0.2588 ± 0.0207 | 0.3031 ± 0.0242 | 0.2926 ± 0.0234 | 0.3945 ± 0.0316 |
| FNN | | 0.3123 ± 0.0250 | 0.2083 ± 0.0167 | 0.1013 ± 0.0081 | 0.1941 ± 0.0155 | 0.1142 ± 0.0091 | 0.0811 ± 0.0065 | 0.3678 ± 0.0294 | 0.3571 ± 0.0286 | 0.1366 ± 0.0109 |
| FAME w/o Bi-dir | | 0.1832 ± 0.0150 | 0.0812 ± 0.0066 | 0.0654 ± 0.0055 | **0.1530** ± 0.0125 | 0.0528 ± 0.0023 | 0.0355 ± 0.0016 | 0.1919 ± 0.0154 | 0.0813 ± 0.0065 | 0.0368 ± 0.0029 |
| FAME w/o MoE | | 0.1870 ± 0.0161 | 0.0828 ± 0.0072 | 0.0663 ± 0.0062 | 0.1578 ± 0.0132 | 0.0538 ± 0.0023 | 0.0362 ± 0.0017 | 0.1972 ± 0.0158 | 0.0856 ± 0.0068 | 0.0374 ± 0.0030 |
| FAME w/o Cross-attn | | 0.1902 ± 0.0158 | 0.0815 ± 0.0078 | 0.0668 ± 0.0064 | 0.1602 ± 0.0143 | 0.0544 ± 0.0026 | 0.0375 ± 0.0018 | 0.1997 ± 0.0160 | 0.0879 ± 0.0070 | 0.0381 ± 0.0030 |
| **FAME** | | **0.1806 ± 0.0152** | **0.0783 ± 0.0064** | **0.0635 ± 0.0056** | 0.1532 ± 0.0120 | **0.0511 ± 0.0023** | **0.0342 ± 0.0017** | **0.1954 ± 0.0156** | **0.0796 ± 0.0064** | **0.0352 ± 0.0028** |

Table 7: Average test-set MSE in simulation (mean $\pm$ standard deviation). The best MSE for each case is highlighted in bold.

| Model | case 4 | case 5 | | | case 6 | case 7 | | case 8 |
| --- | --- | --- | --- | --- | --- | --- | --- | --- |
| | | 0.1 | 0.2 | 0.3 | | 5 | 10 | |
| Basis Expansion (best) | $0.3610 \pm 0.0289$ | $0.3665 \pm 0.0293$ | $0.3944 \pm 0.0316$ | $0.4419 \pm 0.0354$ | $0.3669 \pm 0.0294$ | $0.4501 \pm 0.0360$ | $0.4705 \pm 0.0376$ | $0.5204 \pm 0.0416$ |
| FPCA | $0.3802 \pm 0.0304$ | $0.3879 \pm 0.0310$ | $0.3956 \pm 0.0316$ | $0.4570 \pm 0.0366$ | $0.3844 \pm 0.0308$ | $0.5284 \pm 0.0423$ | $0.5573 \pm 0.0446$ | $0.5737 \pm 0.0459$ |
| Kernel Method | $0.1928 \pm 0.0154$ | $0.1440 \pm 0.0115$ | $0.1919 \pm 0.0154$ | $0.2435 \pm 0.0195$ | $0.2022 \pm 0.0162$ | $0.4659 \pm 0.0373$ | $0.5236 \pm 0.0419$ | $0.5895 \pm 0.0472$ |
| Gaussian Process | $0.2302 \pm 0.0184$ | $0.3079 \pm 0.0246$ | $0.3356 \pm 0.0268$ | $0.3830 \pm 0.0306$ | $0.3434 \pm 0.0275$ | $0.4120 \pm 0.0330$ | $0.4498 \pm 0.0360$ | $0.4762 \pm 0.0381$ |
| FNN | $0.2102 \pm 0.0168$ | $0.1879 \pm 0.0150$ | $0.2250 \pm 0.0180$ | $0.3438 \pm 0.0275$ | $0.1744 \pm 0.0140$ | $0.4801 \pm 0.0384$ | $0.5164 \pm 0.0413$ | $0.5384 \pm 0.0431$ |
| **FAME** | $\mathbf{0.0798 \pm 0.0064}$ | $\mathbf{0.0846 \pm 0.0068}$ | $\mathbf{0.1076 \pm 0.0086}$ | $\mathbf{0.1420 \pm 0.0114}$ | $\mathbf{0.0824 \pm 0.0066}$ | $\mathbf{0.2285 \pm 0.0183}$ | $\mathbf{0.3330 \pm 0.0266}$ | $\mathbf{0.3530 \pm 0.0180}$ |

Table 8: Real-world datasets: average test MSE ($\pm$ standard deviation).

| Model | Hawaii Ocean Salinity | Hawaii Ocean Temp | Human3.6M Walking | Human3.6M Eating | Human3.6M Sitting down | ETDataset Oil Temp |
|---|---|---|---|---|---|---|
| Basis Expansion (best) | $0.0780 \pm 0.0039$ | $0.0014 \pm 0.0001$ | $0.0359 \pm 0.0018$ | $0.0484 \pm 0.0024$ | $0.0122 \pm 0.0006$ | $0.0365 \pm 0.0018$ |
| FPCA | $0.0865 \pm 0.0043$ | $0.0025 \pm 0.0001$ | $0.0373 \pm 0.0019$ | $0.0099 \pm 0.0005$ | $0.0121 \pm 0.0007$ | $0.0355 \pm 0.0018$ |
| Kernel Method | $0.0754 \pm 0.0038$ | $0.0025 \pm 0.0001$ | $0.0373 \pm 0.0019$ | $0.0099 \pm 0.0005$ | $0.0121 \pm 0.0006$ | $0.0355 \pm 0.0016$ |
| Gaussian Process | $0.0931 \pm 0.0047$ | $0.0022 \pm 0.0001$ | $0.0360 \pm 0.0018$ | $0.0075 \pm 0.0004$ | $0.0107 \pm 0.0005$ | $0.0380 \pm 0.0019$ |
| FNN | $0.0766 \pm 0.0038$ | $0.0020 \pm 0.0001$ | $0.0373 \pm 0.0019$ | $0.0099 \pm 0.0005$ | $0.0121 \pm 0.0006$ | $0.0355 \pm 0.0018$ |
| FAME w/o bidir attention | $0.0751 \pm 0.0023$ | $0.00124 \pm 0.00004$ | $0.0327 \pm 0.0010$ | $0.0035 \pm 0.00010$ | $0.0071 \pm 0.0002$ | $0.0264 \pm 0.0008$ |
| FAME w/o MoE | $0.0759 \pm 0.0023$ | $0.00130 \pm 0.00004$ | $0.0332 \pm 0.0010$ | $0.0038 \pm 0.00011$ | $0.0075 \pm 0.00023$ | $0.0271 \pm 0.0008$ |
| FAME w/o cross-attention | $0.0773 \pm 0.0024$ | $0.00145 \pm 0.00005$ | $0.0344 \pm 0.0011$ | $0.0044 \pm 0.00013$ | $0.0083 \pm 0.00025$ | $0.0286 \pm 0.00086$ |
| FAME | $\mathbf{0.0748 \pm 0.0022}$ | $\mathbf{0.0012 \pm 0.00004}$ | $\mathbf{0.0325 \pm 0.0010}$ | $\mathbf{0.0034 \pm 0.00010}$ | $\mathbf{0.0070 \pm 0.00020}$ | $\mathbf{0.0262 \pm 0.00080}$ |

