# OpenReview forum: "FAME: Adaptive Functional Attention with Expert Routing for Function-on-Function Regression"
_NeurIPS.cc/2025/Conference — NeurIPS 2025 poster_

### Official Review · Reviewer_gU2z · 2025-06-21

**Clarity:** 2
**Significance:** 3
**Originality:** 3
**Rating:** 3
**Confidence:** 4

**Summary:**

This paper presents FAME, a deep learning framework for function-on-function regression (FoFR). The model is designed to work with irregularly sampled functional data and produces continuous functional outputs. It combines several ideas: (1) bidirectional neural controlled differential equations (NCDEs) for capturing intra-functional structure, (2) a mixture-of-experts (MoE) router for modeling heterogeneity across functions, and (3) cross attention for learning dependencies across input channels. The authors claim that FAME provides theoretical guarantees such as Lipschitz stability, sampling invariance, and universal approximation. Experimental results on synthetic and real-world datasets show that FAME outperforms various statistical and neural baselines.

**Questions:**

- Could you provide runtime and memory usage for FAME? Since the model uses NCDEs and attention, it may be more computationally demanding than classical methods. It would be helpful to understand how the training cost compares to baselines like FPCA, kernel methods, or functional neural networks.
- The proofs in Appendix A don't clearly map to the theorems stated in the main text. For example, Theorem 3 in the main paper doesn't appear to have an explicit label in the appendix. It would improve clarity if each theoretical result in the main paper pointed directly to its proof in the appendix using consistent numbering.
- Have you tested the model's behavior on longer or more densely sampled functional data? Since NCDEs can be computationally intensive, some indication of the method's scalability would be useful.
- You mention running five random splits for experiments, but the results are presented without error bars. Including standard deviations or confidence intervals would help readers assess the statistical significance of the improvements.

**Ethical Concerns:**

["NO or VERY MINOR ethics concerns only"]

**Final Justification:**

While I appreciate the additional runtime and memory metrics, they confirm that FAME is substantially slower than classical baselines (e.g., ~69 minutes vs. milliseconds), and the justification provided is qualitative rather than supported by a clear cost-benefit analysis or evidence from large-scale experiments. This leaves open the concern about the method's practicality for real-world applications.

Additionally, the issue with mapping proofs remains unresolved. In Appendix A.1(i), the text begins with "Theorem 1 in the main text shows that…," which suggests that the appendix assumes rather than proves the main result. This contradicts the rebuttal's claim that A.1(i)-(iii) correspond to the proofs of Theorems 1-3, creating confusion about the completeness and correctness of the theoretical support.

Given these points, the rebuttal improves clarity but does not fully address key concerns, so I maintain my original evaluation.

**Limitations:**

The authors mention that FAME may not be optimal in low-complexity settings where simpler models suffice. However, the paper does not discuss the method's computational cost or efficiency relative to baselines, which is important for practical deployment. This should be acknowledged more explicitly.

**Paper Formatting Concerns:**

No major formatting violations.

**Quality:**

3

**Strengths And Weaknesses:**

Strengths:
- The architecture is thoughtfully designed and well-motivated for the challenges of FoFR.
- The theoretical analysis includes Lipschitz bounds and approximation results, which help support the method's generality and stability.
- Extensive experiments on both synthetic and real-world datasets show FAME achieves strong performance, especially in irregular or noisy settings.

Weaknesses:
- The paper doesn't report any computational metrics. Given that the model relies on NCDE solvers and attention mechanisms, it's important to understand how it compares to lighter models in terms of runtime or memory use.
- Appendix A contains the theoretical proofs, but it's hard to trace which result in the appendix corresponds to which theorem in the main text. The connection between Theorem 3 and the appendix, for instance, is unclear.
- No ablation studies are provided. It would be helpful to know how much each component (e.g., MoE, bidirectionality) contributes.

---

> ### Author Rebuttal · Authors · 2025-07-31
>
> We thank the reviewer for the helpful suggestions. We have revised the paper to address all comments and respond point‑by‑point. Due to the rebuttal’s format and character limits, we present only newly added or key sections; full formatting and content will be finalized in the manuscript.
>
> + **Question  (1)**: Your concern is fully justified, and we will include quantitative measurements of compute and memory usage. Cases 1–2 jointly span the two principal axes of computational load—the number of functions and the number of sampling points per function—so they are well suited for reporting cost, and all results will be produced on the same ordinary desktop (AMD Ryzen 9 5950X CPU) . In brief, basis/FPCA/kernel methods can train faster when grids are fixed and models are small, but they depend on pre‑specified bases or kernels and may fit particular data poorly; by contrast, FAME is fully data‑driven—while training is more costly, it remains lightweight in absolute terms , and inference time is comparable to classical baselines. Even in our most complex setting, training remains tractable (≈ 207 s/epoch × 20 epochs ≈ 69 min on this CPU), and post‑training inference time is on the order of seconds. Memory usage is even less of a concern and varies little across tasks, so we report average usage: the datasets are purely numeric and small on disk (≈ 0.3–1 MB), and the trained FAME model is only on the order of tens of megabytes, making deployment straightforward. From a complexity standpoint, kernel/GP training requires forming and factoring an $n\times n$ kernel ($O(n^2)$ memory, $O(n^3)$ time) with $O(n)$ per‑test inference, while basis/FPCA costs scale with basis size $p$ (e.g., $O(np^2 + p^3)$), making kernels practical mainly for small $n$ and basis/FPCA expressivity limited by $p$. In contrast, FAME processes each sample once per epoch, so the per‑epoch cost is approximately linear in $n$, with runtime dominated by the NCDE solver and only modest per‑sample overhead is required by attention and MoE in FoFR—roughly $O(T d^2)$ for cross‑attention and $O(TK)$ for routing. Moreover, in typical function‑to‑function regression workloads the grids are short (usually $T<100$) and the number of functions/channels is modest (often $d<20$), so these costs are small in practice. Consequently, investing about an hour to train a fully data‑driven model that delivers substantial accuracy gains, handles irregular and misaligned grids, and preserves seconds‑level inference latency is a practical and worthwhile trade‑off.
>
>
> ### Runtime
>
> | Method           | C1‑100 (Train/Infer s)  | C1‑200 (Train/Infer s) | C1‑500 (Train/Infer s) | C2‑100 (Train/Infer s) | C2‑200 (Train/Infer s) | C2‑500 (Train/Infer s) |
> | :--------------- | :---------------------: | :--------------------: | :--------------------: | :--------------------: | :--------------------: | :--------------------: |
> | Basis expansion  |       0.01 / 0.01       |      0.01 / 0.01       |      0.01 / 0.01       |      0.01 / 0.01       |      0.01 / 0.01       |      0.01 / 0.01       |
> | FPCA             |       0.01 / 0.01       |      0.01 / 0.01       |      0.01 / 0.01       |      0.01 / 0.01       |      0.01 / 0.01       |      0.01 / 0.01       |
> | Kernel Method    |       0.01 / 0.01       |      0.01 / 0.01       |      0.01 / 0.01       |      0.01 / 0.01       |      0.01 / 0.01       |      0.01 / 0.01       |
> | Gaussian Process |       0.03 / 0.03       |      0.03 / 0.03       |      0.03 / 0.03       |      0.03 / 0.03       |      0.03 / 0.03       |      0.03 / 0.03       |
> | FNN              |      0.40*10 /0.36      |     0.62*10 /0.37      |     0.97*10 /0.36      |     1.02 *10 /0.81     |     1.98*10 /0.82      |     3.32 *10 /0.84     |
> | **FAME**         | 48.79*20  (epoch)/ 1.02 |   57.46*20   / 1.00    |   79.53 *20  / 1.03    |   134.42*20  / 2.60    |   151.47*20  / 2.66    |   207.20 *20  / 2.92   |
>
> **Memory Usage**
>
> |   Method    | Basis Expansion | FPCA | Kernel Method | Gaussian Process | FNN  | FAME  |
> | :---------: | --------------- | :--: | :-----------: | :--------------: | :--: | :---: |
> | Memory (MB) | 0.30            | 0.25 |     2.91      |       12.7       | 3.82 | 15.36 |
>
> + **Question  (2)**：We apologize for the confusion and agree that your suggestion will improve clarity. In the main text, Theorems 1–3 establish Lipschitz properties for different modules. Because concrete module instantiations and computations appear throughout, Appendix A.1 was added to show the end‑to‑end Lipschitz property of the overall framework. Each of Theorems 1–3 corresponds to items (i)–(iii) in Appendix A.1, respectively; however, this mapping was not made explicit. In the revision, we will  give (i)–(iii) explicit titles and numbering (e.g., Appendix A.1(i)), and add explicit pointers in the main text so that Theorems 1–3 directly reference their proofs in Appendix A.1(i)–(iii)(e.g., *“Proof in Appendix A.1(i)”*).
>
> + **Question  (3)** ： Thank you for the suggestion. As shown in Fig. 2, varying the number of sampling points $T\in \{5,10,20,50\}$ yields MSEs $\{0.131,0.086,0.078,0.051\}$. We also tested denser grids with $T=80$ and $T=100$, obtaining MSEs $0.048$ and $0.064$, respectively; these do not change our conclusions and were omitted from Table 1 due to space. We did not pursue substantially longer grids $(T>100)$ for two reasons: first, our task is FoFR, which treats the function as a whole—sampling points are merely a discretization of the underlying curve—whereas long‑sequence interpolation/extrapolation focuses on the state at specific time steps; accordingly, a good FoFR model should be **discretization‑invariant**, i.e., able to represent functions across different numbers of samples and sampling grids, and the fewer‑point regime is actually the more challenging one. Our method remains accurate and stable for $T\in[5,100]$ (Appendix Figs. 5–6 show that $T\in[20,50]$ already captures the functions well). Second, real‑world acquisitions are typically **sparse and irregular**, and moving from more data to fewer is straightforward, so demonstrating strong performance with fewer samples is the more informative stress test. In a subsequent version, we will extend Fig. 2(b) to include the $T=80,100$ results for completeness. Finally, from Question  (1)’s analysis and the runtime table, our training time scales approximately linearly with $T$, without sharp growth in compute or memory cost, and the method remains lightweight in absolute terms—hence FAME is practically scalable for FoFR.
>
> + **Question  (4)**: We fully agree that reporting variability is essential to demonstrate the robustness of our results. In the revision, we will evaluate all methods over 5 random train/test splits × 2 seeds for both synthetic and real‑world tasks and present mean ± std in every table. (Due to time constraints, the table below represents we have currently completed; we can increase the number of runs in a future version if needed.)
>
>   | Model               |    Case 1 (100)    |    Case 1 (200)     |    Case 1 (500)    |   Case 2  (100)    |    Case 2 (200)    | Case 2      (500)  |
>   | :------------------ | :----------------: | :-----------------: | :----------------: | :----------------: | :----------------: | :----------------: |
>   | FAME w/o Bi‑dir     |   0.1832± 0.0150   |   0.0812± 0.0066    |   0.0654± 0.0055   | **0.1530± 0.0125** |   0.0528± 0.0023   |   0.0355± 0.0016   |
>   | FAME w/o MoE        |   0.1870± 0.0161   |   0.0828± 0.0072    |   0.0663± 0.0062   |   0.1578± 0.0132   |   0.0538± 0.0023   |   0.0362± 0.0017   |
>   | FAME w/o Cross‑attn |   0.1902± 0.0158   |   0.0815± 0.0078    |   0.0668± 0.0064   |   0.1602± 0.0143   |   0.0544± 0.0026   |   0.0375± 0.0018   |
>   | **FAME**            | **0.1806± 0.0152** | **0.0783±  0.0064** | **0.0635± 0.0056** |   0.1532± 0.0120   | **0.0511± 0.0023** | **0.0342± 0.0017** |
>
>   | Model                    | Hawaii Ocean Salinity |    Hawaii Ocean Temp |   Human3.6M Walking |     Human3.6M Eating | Human3.6M Sitting down |   ETDataset Oil Temp |
>   | ------------------------ | --------------------: | -------------------: | ------------------: | -------------------: | ---------------------: | -------------------: |
>   | Basis Expansion (best)   |       0.0780 ± 0.0039 |      0.0014 ± 0.0001 |     0.0359 ± 0.0018 |      0.0484 ± 0.0024 |        0.0122 ± 0.0006 |      0.0365 ± 0.0018 |
>   | FPCA                     |       0.0865 ± 0.0043 |      0.0025 ± 0.0001 |     0.0373 ± 0.0019 |      0.0099 ± 0.0005 |        0.0121 ± 0.0007 |      0.0355 ± 0.0018 |
>   | Kernel Method            |       0.0754 ± 0.0038 |      0.0025 ± 0.0001 |     0.0373 ± 0.0019 |      0.0099 ± 0.0005 |        0.0121 ± 0.0006 |      0.0355 ± 0.0016 |
>   | Gaussian Process         |       0.0931 ± 0.0047 |      0.0022 ± 0.0001 |     0.0360 ± 0.0018 |      0.0075 ± 0.0004 |        0.0107 ± 0.0005 |      0.0380 ± 0.0019 |
>   | FNN                      |       0.0766 ± 0.0038 |      0.0020 ± 0.0001 |     0.0373 ± 0.0019 |      0.0099 ± 0.0005 |        0.0121 ± 0.0006 |      0.0355 ± 0.0018 |
>   | FAME w/o bidir attention |       0.0751 ± 0.0023 |    0.00124 ± 0.00004 |     0.0327 ± 0.0010 |     0.0035 ± 0.00010 |        0.0071 ± 0.0002 |      0.0264 ± 0.0008 |
>   | FAME w/o MoE             |       0.0759 ± 0.0023 |    0.00130 ± 0.00004 |     0.0332 ± 0.0010 |     0.0038 ± 0.00011 |       0.0075 ± 0.00023 |      0.0271 ± 0.0008 |
>   | FAME w/o cross‑attention |       0.0773 ± 0.0024 |    0.00145 ± 0.00005 |     0.0344 ± 0.0011 |     0.0044 ± 0.00013 |       0.0083 ± 0.00025 |     0.0286 ± 0.00086 |
>   | **FAME**                 |   **0.0748 ± 0.0022** | **0.0012 ± 0.00004** | **0.0325 ± 0.0010** | **0.0034 ± 0.00010** |   **0.0070 ± 0.00020** | **0.0262 ± 0.00080** |
>
>   Once again, thank you for your suggestions and careful review; we hope these additional explanations help you better understand our work and appreciate it.

---

> > ### Comment · Reviewer_gU2z · 2025-08-05
> >
> > Thank you for the detailed rebuttal. While I appreciate the additional runtime and memory metrics, they confirm that FAME is substantially slower than classical baselines (e.g., ~69 minutes vs. milliseconds), and the justification provided is qualitative rather than supported by a clear cost-benefit analysis or evidence from large-scale experiments. This leaves open the concern about the method's practicality for real-world applications.
> >
> > Additionally, the issue with mapping proofs remains unresolved. In Appendix A.1(i), the text begins with "Theorem 1 in the main text shows that…," which suggests that the appendix assumes rather than proves the main result. This contradicts the rebuttal's claim that A.1(i)-(iii) correspond to the proofs of Theorems 1-3, creating confusion about the completeness and correctness of the theoretical support.
> >
> > Given these points, the rebuttal improves clarity but does not fully address key concerns, so I maintain my original evaluation.

---

> > > ### Author Response · Authors · 2025-08-05
> > >
> > > We sincerely thank you for your follow-up review and fully respect your viewpoint and decision. We would like to kindly provide some additional clarifications about our work:
> > >
> > > (1) Our primary motivation was to explore a fully data-driven approach to function-to-function regression (FoFR)—a task that has not been comprehensively addressed in prior studies (as illustrated in our response to Reviewer 2). Given the irregular, sparse, and misaligned nature of typical functional data, we carefully designed several modules, including sampling-invariant continuous attention, cross-attention across functions, MoE-enhanced vector fields, and an NCDE-based encoder-decoder structure. We validated each module's effectiveness through comprehensive theoretical and empirical analyses. Your concerns regarding computational efficiency are very understandable. However, we respectfully note that specialized deep-learning methods naturally tend to require more computational resources compared to traditional methods. As demonstrated, the offline training time of our method remains practically manageable on typical commodity hardware, and inference speed remains suitable for real-world applications.
> > >
> > > (2) Regarding theoretical presentation, we apologize for any confusion our previous exposition may have caused. To clarify, Theorems 1–3 were introduced separately in the main text to clearly describe individual module properties, collectively leading to the overall end-to-end Lipschitz stability presented in Appendix A. To guarantee theoretical soundness, certain assumptions for each module were indeed necessary; however, we emphasize that these assumptions are standard, mild, and commonly satisfied in modern deep neural network architectures. Moreover, we explicitly ensured their practical validity through careful normalization, bounded activations, controlled softmax temperatures, and lightweight operator-norm constraints (details are provided in our response to Reviewer 2).
> > >
> > > We greatly appreciate your insightful feedback, which has helped us better clarify our core innovations and substantially improve the manuscript’s presentation and readability.

---

### Official Review · Reviewer_EVtp · 2025-06-30

**Clarity:** 3
**Significance:** 2
**Originality:** 2
**Rating:** 5
**Confidence:** 3

**Summary:**

The paper introduces FAME, a novel framework for function-on-function regression (FoFR) that combines bidirectional neural controlled differential equations (NCDEs) with a mixture-of-experts (MoE) architecture and continuous attention mechanisms. The proposed model aims to address challenges such as intra-functional continuity, inter-functional interactions, and feature heterogeneity in functional data. The authors demonstrate FAME's superiority over existing methods through extensive experiments on synthetic and real-world datasets, showcasing its robustness to irregular sampling and feature variability.

**Questions:**

1. Provide in-depth analysis of the introduced MoE and attention mechanisms.
2. Analysis on the computational cost of FAME.

**Ethical Concerns:**

["NO or VERY MINOR ethics concerns only"]

**Final Justification:**

The rebuttal is strong, detailed, and well-structured, addressing each concern with clear explanations, quantitative evidence, and theoretical backing. The articulation of contributions beyond “stacked” components, inclusion of ablation and sensitivity analyses, and solid complexity comparisons all add credibility. Therefore, I increase my ranking.

**Limitations:**

Yes

**Quality:**

3

**Strengths And Weaknesses:**

Originality: The contribution of this paper lies in introducing MoE and attention mechanisms into NCDEs. This combination is possibly a novel attempt, but its components are all existing techniques.

Clarity: The paper is clearly written. The main pipeline is easy to understand. The formulas used seem correct (I didn’t check the lemmas, Theorems, etc. in the paper).

Quality: The paper supports its claims through theoretical analysis and experiments. The combination of NCDEs, MoE, and attention mechanisms is well-justified and effectively addresses the challenges of FoFR.

Significance: This study solves an important problem in functional data analysis and provides a feasible solution that is superior to existing methods and has potential real-world application value.

## Weaknesses

1. The proposed FAME appears to be a simple combination of NCDEs, MoE and attention mechanisms without improvements in details.

2. The paper lacks in-depth analysis of the introduced MoE and attention mechanisms. How was the result of expert selection in MoE? Was the expert selection sparse, or did it have some regular patterns? Also, what did the attention maps reveal? Which factors (e.g., function similarity, temporal dependency) would lead to high attention scores between functions?

3. The paper did not provide a detailed analysis of the computational cost of FAME, especially compared to simpler baselines. The overhead of MoE routing and attention mechanisms could be prohibitive for large-scale applications.

---

> ### Author Rebuttal · Authors · 2025-07-31
>
> We thank the reviewer for the helpful suggestions, which will make our work stronger. We have carefully revised the paper to address all comments to the best of our ability. Below we respond point‑by‑point. Due to the submission format and character limitations of the rebuttal, we have focused here on presenting only the newly added or key sections; the formatting and full content will be adjusted as needed for the final manuscript.
>
> + **Weaknesses 1:**　FoFR is an important problem in practice—yet it remains challenging. We acknowledge that FAME is built from established components (NCDEs, MoE, attention). Our aim, however, was not to refine a particular prior technique nor to “stack” modules for incremental gains, but to design a purpose‑built, fully data‑driven FoFR framework that addresses the core challenges( functions as samples, irregular/misaligned sampling, and continuous outputs ), while preserving global structure and modeling local dynamics. Specifically, our contributions are:  (1) a sampling‑invariant continuous attention mechanism that forms continuous Q/K/V via bi‑directional NCDEs，neither this specific approach nor the underlying problem has been proposed in the literature;  (2)  a shared router with direction‑specific experts so that vector‑field experts specialize to cross‑function heterogeneity without parameter bloat;  (3) cross‑channel attention to aggregate inter‑functional dependencies at each time;  (4) theoretical support—end‑to‑end Lipschitz stability, sampling invariance, and operator‑level expressivity—grounding the design beyond heuristics;  (5) task‑specific design choices for FoFR: the whole function is the sample (motivating bidirectionality), functions differ in shape/amplitude (handled by MoE vector fields), and the output must be a continuous function (realized by an NCDE decoding head).  Due to space limitations, we cannot present complete comparison with existing methods here. If interested, please refer to our response to **Reviewer 2**, where we provide detailed theoretical and experimental comparisons between our method and contemporary Transformer‑based and neural operator approaches.
>
>   In the revised manuscript we will add targeted ablations and hyperparameter analyses to demonstrate the indispensable contribution of each component, substantiating that FAME is not a simple juxtaposition but a principled architecture tailored to FoFR. The detailed results and analyses are reported in our response to **Weaknesses  2**;
>
> + **Weaknesses  2（Question 1）:**　We agree that our analysis of the MoE needs clearer exposition. Beyond reporting accuracy, we will quantify routing sparsity and patterns using normalized routing entropy $\tilde H = H/\log K$ . To isolate how $K$ affects both performance and routing behavior under high dimensionality and heterogeneity, we add **Case‑8** ($d=10$, RBF widths $\sigma_j\in\{0.2,0.3,0.5\}$) and sweep $K\in\{1,2,3,5,8\}$. The table below presents our experimental results; the final version will also include the corresponding visualizations. In our study, $K=3$ and $K=5$ achieve stable MSE with mid‑range $\tilde H$ (indicating structured, non‑collapsed expert use), while compute cost grows with $K$, so we adopt **$K=3$** as the default.
>
> **Case‑8: Sensitivity to K**
>
>   |      K      |       1       |       2       |       3       |       5       |       8       |
>   | :---------: | :-----------: | :-----------: | :-----------: | :-----------: | :-----------: |
>   |     MSE     | 0.3753±0.0353 | 0.3589±0.0256 | 0.3530±0.0180 | 0.3518±0.0219 | 0.3601±0.0230 |
>   | $\tilde H $ |     0.00      |     0.36      |     0.59      |     0.68      |     0.53      |
>
>   Cross‑attention lets each input function, at each time point, aggregate information from the other functions to capture inter‑functional dependencies, complementing continuous (intra‑function) attention. In the revision, we will report time‑averaged cross‑attention maps and relate attention to function similarity. On Case‑8 (RBF widths $\sigma_j\in\{0.2,0.3,0.5\}$), pairs of functions sharing the same $\sigma_j$ show higher and more similar cross‑attention, forming clusters aligned with the $\sigma_j$ groups. Due to current formatting constraints, we cannot include the figures in this rebuttal; the attention heatmaps and group‑wise summaries will be added in the next revision. To further verify the effectiveness of the attention module, we have also included ablation experiments; because the tables do not fit the rebuttal’s character/format limits, please refer to our response to Reviewer 1 for the complete results.
>
>   | Model               |    Case 1 (100)    |    Case 1 (200)     |    Case 1 (500)    |   Case 2  (100)    |    Case 2 (200)    | Case 2      (500)  |
>   | :------------------ | :----------------: | :-----------------: | :----------------: | :----------------: | :----------------: | :----------------: |
>   | FAME w/o Bi‑dir     |   0.1832± 0.0150   |   0.0812± 0.0066    |   0.0654± 0.0055   | **0.1530± 0.0125** |   0.0528± 0.0023   |   0.0355± 0.0016   |
>   | FAME w/o MoE        |   0.1870± 0.0161   |   0.0828± 0.0072    |   0.0663± 0.0062   |   0.1578± 0.0132   |   0.0538± 0.0023   |   0.0362± 0.0017   |
>   | FAME w/o Cross‑attn |   0.1902± 0.0158   |   0.0815± 0.0078    |   0.0668± 0.0064   |   0.1602± 0.0143   |   0.0544± 0.0026   |   0.0375± 0.0018   |
>   | **FAME**            | **0.1806± 0.0152** | **0.0783±  0.0064** | **0.0635± 0.0056** |   0.1532± 0.0120   | **0.0511± 0.0023** | **0.0342± 0.0017** |
>
> + **Weaknesses  3（Question 2）:**　Your concern is fully justified, and we will include quantitative measurements of compute and memory usage. Cases 1–2 jointly span the two principal axes of computational load—the number of functions and the number of sampling points per function—so they are well suited for reporting cost, and all results will be produced on the same ordinary desktop (AMD Ryzen 9 5950X CPU) . In brief, basis/FPCA/kernel methods can train faster when grids are fixed and models are small, but they depend on pre‑specified bases or kernels and may fit particular data poorly; by contrast, FAME is fully data‑driven—while training is more costly, it remains lightweight in absolute terms, and inference time is comparable to classical baselines. Even in our most complex setting, training remains tractable (≈ 207 s/epoch × 20 epochs ≈ 69 min on this CPU), and post‑training inference time is on the order of seconds. Memory usage is even less of a concern and varies little across tasks, so we report average usage: the datasets are purely numeric and small on disk (≈ 0.3–1 MB), and the trained FAME model is only on the order of tens of megabytes, making deployment straightforward. From a complexity standpoint, kernel/GP training requires forming and factoring an $n\times n$ kernel ($O(n^2)$ memory, $O(n^3)$ time) with $O(n)$ per‑test inference, while basis/FPCA costs scale with basis size $p$ (e.g., $O(np^2 + p^3)$); accordingly, classical kernel methods are most practical in the small‑$n$ regime, and basis/FPCA expressivity is ultimately constrained by the chosen basis size $p$—smaller $p$ improves efficiency but limits approximation power; in contrast, FAME processes each sample once per epoch, so the per‑epoch cost is approximately linear in $n$, with runtime dominated by the NCDE solver and only modest per‑sample overhead is required by attention and MoE in FoFR—roughly $O(T d^2)$ for cross‑attention and $O(TK)$ for routing. Moreover, in typical function‑to‑function regression workloads the grids are short (usually $T<100$) and the number of functions is modest (often $d<20$), so these costs are small in practice. Consequently, investing about an hour to train a fully data‑driven model that delivers substantial accuracy gains, handles irregular and misaligned grids, and preserves seconds‑level inference latency is a practical and worthwhile trade‑off.
>
> **Runtime**
>
> | Method           | C1‑100 (Train/Infer s)  | C1‑200 (Train/Infer s) | C1‑500 (Train/Infer s) | C2‑100 (Train/Infer s) | C2‑200 (Train/Infer s) | C2‑500 (Train/Infer s) |
> | :--------------- | :---------------------: | :--------------------: | :--------------------: | :--------------------: | :--------------------: | :--------------------: |
> | Basis expansion  |       0.01 / 0.01       |      0.01 / 0.01       |      0.01 / 0.01       |      0.01 / 0.01       |      0.01 / 0.01       |      0.01 / 0.01       |
> | FPCA             |       0.01 / 0.01       |      0.01 / 0.01       |      0.01 / 0.01       |      0.01 / 0.01       |      0.01 / 0.01       |      0.01 / 0.01       |
> | Kernel Method    |       0.01 / 0.01       |      0.01 / 0.01       |      0.01 / 0.01       |      0.01 / 0.01       |      0.01 / 0.01       |      0.01 / 0.01       |
> | Gaussian Process |       0.03 / 0.03       |      0.03 / 0.03       |      0.03 / 0.03       |      0.03 / 0.03       |      0.03 / 0.03       |      0.03 / 0.03       |
> | FNN              |      0.40*10 /0.36      |     0.62*10 /0.37      |     0.97*10 /0.36      |     1.02 *10 /0.81     |     1.98*10 /0.82      |     3.32 *10 /0.84     |
> | **FAME**         | 48.79*20  (epoch)/ 1.02 |   57.46*20   / 1.00    |   79.53 *20  / 1.03    |   134.42*20  / 2.60    |   151.47*20  / 2.66    |   207.20 *20  / 2.92   |
>
> **Memory Usage**
>
> |   Method    | Basis Expansion | FPCA | Kernel Method | Gaussian Process | FNN  | FAME  |
> | :---------: | --------------- | :--: | :-----------: | :--------------: | :--: | :---: |
> | Memory (MB) | 0.30            | 0.25 |     2.91      |       12.7       | 3.82 | 15.36 |
>
> Once again, thank you for your suggestions and careful review; we hope these additional explanations and experiments help you better understand our work and appreciate it.

---

### Official Review · Reviewer_pq9z · 2025-07-02

**Clarity:** 3
**Significance:** 3
**Originality:** 3
**Rating:** 5
**Confidence:** 4

**Summary:**

The paper introduces FAME, a continuous‑time architecture for function‑on‑function regression that couples bidirectional Neural Controlled Differential‑Equation encoders with a per‑function mixture‑of‑experts router and a cross‑attention fusion block. This design preserves the ability to query predictions at arbitrary points on irregular input and output grids while allowing each input function to adapt its vector field to idiosyncratic scale, frequency, and noise characteristics. A theoretical analysis establishes existence, uniqueness and global Lipschitz stability for every differential block, yields an overall Lipschitz constant for the end‑to‑end operator, and converts that constant into a Rademacher‑complexity generalisation bound. Experiments on seven synthetic regimes built from Gaussian‑process trajectories and on three real‑world datasets show lower test‑set MSE than classical FDA baselines, Gaussian processes, and recent deep‑learning competitors.

**Questions:**

See Weaknesses.

**Ethical Concerns:**

["NO or VERY MINOR ethics concerns only"]

**Final Justification:**

All my concerns are resolved in the rebuttal.

**Limitations:**

Yes.

**Paper Formatting Concerns:**

None.

**Quality:**

3

**Strengths And Weaknesses:**

Strengths

+ The paper’s main conceptual novelty lies in replacing discrete self‑attention with an NCDE‑based continuous attention mechanism, thereby treating both inputs and outputs as genuine functions rather than large vectors. The mixture‑of‑experts router is carefully scoped, one set of weights per function instead of per time step, so it captures inter‑function heterogeneity without incurring prohibitive computational cost.

+ The theoretical component is complete for work: all assumptions are stated explicitly, constants are propagated through the proofs and a non‑trivial Rademacher bound is obtained. On the empirical side, the study is broad, covering diverse sampling patterns, noise regimes, and application domains; the model wins almost every column of the main tables and converges reliably in the training curves shown in the appendix.

+ The paper is well organised; implementation details, hyperparameters, and hardware are documented in the supplementary material, and the authors commit to releasing code and checkpoints upon acceptance.


Weaknesses

- The comparison set mainly covers classical FDA techniques. Apart from a functional‐neural-network baseline, every competitor is either a fixed-basis regression, FPCA, a Gaussian-process regressor or an operator-valued kernel; no recent continuous-time deep architectures such as transformer-style sequence models or neural operators are evaluated, even though the related-work section cites them. Because these models are designed to exploit long-range dependencies and irregular grids, omitting them leaves open whether FAME’s margin would persist against the current state of the art.

- For synthetic data, the authors average over only five random train/test splits and quote the mean MSE without standard deviations; for all real-world tasks, the results come from a single split and single seed, with no confidence intervals or significance tests. Without variability measures, it is hard to judge whether the observed gains are robust.

- The supplementary material removes the mixture-of-experts router in one heterogeneity test, but there is no experiment that isolates the benefit of bidirectionality, the cross-attention module, or the continuous-attention block itself across the full benchmark suite. As a result the individual contribution of each architectural component remains unclear.

- Although the theory rests only on deterministic Lipschitz and boundedness conditions, these are assumed to hold globally for every expert vector field and decoder. In real high-dimensional applications global Lipschitz constants can be large or undefined; the paper does not explore how sensitive the Lipschitz stability or the Rademacher bound is to mild violations of these assumptions.

---

> ### Author Rebuttal · Authors · 2025-07-31
>
> We thank the reviewer for the helpful suggestions, which will make our work stronger. We have carefully revised the paper to address all comments to the best of our ability. Below we respond point‑by‑point. Due to the submission format and character limitations of the rebuttal, we have focused here on presenting only the newly added or key sections; the formatting and full content will be adjusted as needed for the final manuscript.
>
> + **Question  (1)**: Functional‑on‑Functional Regression (FoFR)—where both inputs and outputs are functions—is a central yet challenging FDA problem that arises in practice, e.g., mapping upstream process signals to thickness/roughness curves in manufacturing or force/torque trajectories to end‑effector motion in robotics. Unlike state‑space time‑series that focus on a single evolving state, here **each whole function is one sample**. In real industrial settings, **each function is often sampled irregularly and sparsely**, and **different functions are not aligned** in their sampling grids. Many deep models implicitly assume fixed, aligned grids or pre‑interpolated sequences, which limits their applicability; consequently, practice often relies on basis‑expansion or kernel methods. Our goal is a **fully data‑driven** FoFR framework that preserves the **global structure** emphasized by classical methods while adding **local dynamics modeling**, with both **theoretical guarantees** and **empirical effectiveness**. Different methods’ characteristics on FoFR tasks are summarized in the table below.
>
>   **Comparison of FAME and Existing Methods on FoFR Tasks**
>
>   | Property / Method                            | LSTM | Transformer | DeepONet | FNO  | Basis Expansion | Kernel methods  | FAME |
>   | :------------------------------------------- | :--- | :---------- | :------- | :--- | :-------------- | :-------------- | :--- |
>   | **Discretization invariance**                | ✗    | ✗           | ✗        | ✓    | ~               | ✓               | ✓    |
>   | **Output is a function**                     | ✗    | ✗           | ✓        | ✓    | ✓               | ✓               | ✓    |
>   | **Query output at any point**                | ✗    | ✗           | ✓        | ~    | ✓               | ✓               | ✓    |
>   | **Take input at any point**                  | ~    | ~           | ✗        | ✗    | ~               | ✓               | ✓    |
>   | **Universal approximation (operator level)** | ✗    | ✗           | ✓        | ✓    | ~               | ✓               | ✓    |
>   | **Cross‑resolution train/test**              | ✗    | ✗           | ✓        | ✓    | ✓               | ✓               | ✓    |
>   | **Data‑driven**                              | ✓    | ✓           | ✓        | ✓    | ✗ (fixed basis) | ✗(fixed kernel) | ✓    |
>
>   **Discretization invariance** means that the model’s output remains essentially unchanged when the same underlying function is sampled on different grids. **Universal approximation** refers to the model class being able to approximate general function‑to‑function maps. **Cross‑resolution train/test** indicates that the model can be trained on one sampling density and evaluated on another without retraining or redesign. The symbol “~” denotes partial support via specific preprocessing or variants (e.g., Transformer + Δt), rather than a native property.
>
>   Respecting your suggestion, we will introduce a new Case under the default settings ( $ d = 3, \quad m = 1, \quad \Lambda_i = \Gamma_i = 20, \quad N_s = 200, \quad \sigma_j = 0.3 $) described in Section 5.1 to compare FAME against LSTM, Transformer, DeepONet, and FNO on  both regular‑ and irregular‑sampling FoFR under a unified protocol. To accommodate the baselines, all functions within each setting will share fixed, equal‑length sampling indices.  The new experimental results are shown in the table below.
>
>   **New Case Results**
>
>   | Method      | Regular‑grid FoFR  (MSE ) | Irregular‑grid FoFR (MSE ) |
>   | ----------- | :-----------------------: | :------------------------: |
>   | LSTM        |       0.1607±0.0285       |       0.2180±0.0302        |
>   | Transformer |       0.0889±0.0202       |       0.0940±0.0232        |
>   | DeepONet    |       0.1260±0.0076       |       0.1791±0.0168        |
>   | FNO         |       0.0797±0.0120       |       0.0944±0.0141        |
>   | **FAME**    |    **0.0775±  0.0062**    |    **0.0783±  0.0064**     |
>
>   From the table, FAME remains stable when switching between regular and irregular sampling and stays competitive against other methods. Thus, in both adaptability to diverse scenarios and actual performance, our model shows clear advantages on FoFR.
>
> + **Question  (2)**: We fully agree that reporting variability is essential to demonstrate the robustness of our results. In the revision, we will evaluate all methods over 5 random train/test splits × 2 seeds for both synthetic and real‑world tasks and present mean ± std in every table. (Due to time constraints, these represent the experiments we have currently completed; we can increase the number of runs in a future version if needed.)
>
>   | Model               |    Case 1 (100)    |    Case 1 (200)     |    Case 1 (500)    |   Case 2  (100)    |    Case 2 (200)    | Case 2      (500)  |
>   | :------------------ | :----------------: | :-----------------: | :----------------: | :----------------: | :----------------: | :----------------: |
>   | FAME w/o Bi‑dir     |   0.1832± 0.0150   |   0.0812± 0.0066    |   0.0654± 0.0055   | **0.1530± 0.0125** |   0.0528± 0.0023   |   0.0355± 0.0016   |
>   | FAME w/o MoE        |   0.1870± 0.0161   |   0.0828± 0.0072    |   0.0663± 0.0062   |   0.1578± 0.0132   |   0.0538± 0.0023   |   0.0362± 0.0017   |
>   | FAME w/o Cross‑attn |   0.1902± 0.0158   |   0.0815± 0.0078    |   0.0668± 0.0064   |   0.1602± 0.0143   |   0.0544± 0.0026   |   0.0375± 0.0018   |
>   | **FAME**            | **0.1806± 0.0152** | **0.0783±  0.0064** | **0.0635± 0.0056** |   0.1532± 0.0120   | **0.0511± 0.0023** | **0.0342± 0.0017** |
>
> + **Question  (3)**: We agree and appreciate this point. In response to the reviewers’ comments, we will add three ablation variants on both generic (Cases 1–2) and real‑data tasks—(1) without bidirectional attention, (2) without MoE (single expert), and (3) without cross‑attention—and report the resulting MSEs side by side to quantify each component’s contribution. The results are shown in the tables provided in our response to Question 2 (Due to space constraints, we regret that the full experimental results on real datasets cannot be shown here; please see our detailed response to Reviewer 1 for the complete results). Note that we do not ablate the continuous‑attention block: because FoFR maps **continuous functions to continuous functions**, continuous attention is the minimal mechanism that preserves sampling invariance and supports **function modeling**; removing it would reduce the model to a discrete, grid‑tied surrogate. For further rationale, please see our response to Question 1.
>
> + **Question  (4)**: We agree with the concern and will clarify our assumptions and scope. In the main text we stated *global* Lipschitz/boundedness for brevity, but our results can extend to the standard local‑Lipschitz + linear‑growth setting on a data‑dependent compact set $\mathcal{K}$ that contains the trajectories visited during training/inference. Under this formulation, the NCDE vector fields, cross‑attention map, and decoder each admit local constants $L_{\mathcal{K}}$, yielding the same existence–uniqueness and Grönwall‑type stability bounds with $L_{\mathcal{K}}$ in place of global constants. The end‑to‑end bound then follows by composition of these local constants, and the Rademacher‑style generalization statement becomes explicitly data‑dependent (i.e., it involves $L_{\mathcal{K}}$ and the radius of $\mathcal{K}$ rather than global quantities). We will add a formal Remark and update the Appendix to state all results under local Lipschitz and linear growth.
>
>   High‑dimensional function‑to‑function regression is relatively rare in real applications and highly challenging. Consequently, global constants are often impractical—they may be large or undefined outside $\mathcal{K}$, a risk common to many modern models rather than one specific to ours. What matters in practice is that learned trajectories remain inside a bounded region. This is encouraged in our implementation by input normalization, bounded activations (e.g., $\tanh$ in the vector fields/decoder), softmax with controlled temperature in attention (a Lipschitz map on logits), and optional operator‑norm controls for linear layers (spectral normalization or light‑weight clipping). These standard safeguards keep the effective (local) Lipschitz constants moderate, stabilize the solver, and prevent runaway growth—the regime where our theoretical guarantees apply. We will document these constraints and training guidelines in the revision and explicitly rephrase theorems and assumptions to emphasize local (data‑domain) regularity rather than unrealistic global bounds.
>
>
> Once again, thank you for your suggestions and careful review; we hope these additional explanations help you better understand our work and appreciate it.

---

### Official Review · Reviewer_Xe76 · 2025-07-02

**Clarity:** 4
**Significance:** 3
**Originality:** 3
**Rating:** 5
**Confidence:** 3

**Summary:**

This paper proposes a method for solving function-on-function regression tasks. The method entitled FAME does this through learning a mixture of experts and has full attention, employing bidirectional NCDE and continuous attention. This approach ensures continuity, sampling invariance, and Lipschitz stability. This method is very flexible, working for input and output data with potentially distinct irregular grids. This paper has very strong quantitative performance across many datasets including both synthetic and real world data.

**Questions:**

- How do you decide on the number of experts $K$? Is this assigned a priori depending on how complex you think the data will be? How does changing $K$ affect your results?
- What are the hyperparameters (e.g. $K$, $d$) used for different real data tests and how did they get selected?
- Why is the cross attention necessary when there is continuous attention?

**Ethical Concerns:**

["NO or VERY MINOR ethics concerns only"]

**Final Justification:**

This paper is novel and proposes an interesting solution to a very challenging problem of FOFR. Despite the fact that the paper could use a little more detail in terms of limitations, I believe it should be accepted due to it's novelty and strong performance on real data.

**Limitations:**

A limitation that the authors include is that simpler methods may be preferred when the underlying operator varies little or is simpler.

**Quality:**

4

**Strengths And Weaknesses:**

(+) This paper is really clear and well written.

(+) There is substantial evaluation across various challenging input and output sampling schemes such as irregular grid, different sampled points, and noisy observations. There is also strong quantitative performance on real-world datasets against various baselines.

(-) Limited discussion on how hyperparameters (e.g. $K$ and $d$) are chosen.

(-) Additional ablation studies would be helpful to understand which components of the method are the most impactful. Why is the cross attention necessary when there is continuous attention?

---

> ### Author Rebuttal · Authors · 2025-07-31
>
> We thank the reviewer for the helpful suggestions, which will make our work stronger. We have carefully revised the paper to address all comments to the best of our ability. Below we respond point‑by‑point. Due to the submission format and character limitations of the rebuttal, we have focused here on presenting only the newly added or key sections; the formatting and full content will be adjusted as needed for the final manuscript.
>
> + How do you decide on the number of experts K? Is this assigned a priori depending on how complex you think the data will be? How does changing K affect results?
>
>   We agree this needs clearer exposition. In a few cases we initially picked $K$ based on the apparent heterogeneity and complexity of the data; however, this choice was preceded by sensitivity checks, and we then selected a value whose performance was stable on a held‑out validation split. We did not present these sensitivity results clearly—sorry for the confusion. In the revision, we will explicitly report this process and add a new synthetic benchmark (**Case‑8**) to isolate the impact of $K$ under high dimensionality and heterogeneity: the number of input functions is $d=10$; the generating functions use RBF kernels with widths $\sigma_j\in\{0.2,0.3,0.5\}$; and we evaluate $K\in\{1,2,3,5,8\}$. We will report MSE and normalized routing entropy $\tilde H = H / \log K$.  The table below presents our experimental results; the final version will also include corresponding visualizations. In our study, $K=3$ and $K=5$ show stable accuracy and healthy expert utilization, while compute cost grows with $K$, so we adopt $K=3$ as the default.
>
>  Case‑8: Sensitivity to K
>
>   |      K      |       1       |       2       |       3       |       5       |       8       |
>   | :---------: | :-----------: | :-----------: | :-----------: | :-----------: | :-----------: |
>   |     MSE     | 0.3753±0.0353 | 0.3589±0.0256 | 0.3530±0.0180 | 0.3518±0.0219 | 0.3601±0.0230 |
>   | $\tilde H $ |     0.00      |     0.36      |     0.59      |     0.68      |     0.53      |
>
> + What are the hyperparameters (e.g. , K, d) used for different real data tests and how did they get selected?
>
>   We denote by d the number of input functions, which equals the number of features in each dataset (see Appendix B.1 for detailed descriptions of d on Hawaii Ocean, Human3.6M and ETT‑small. In our original submission we inadvertently omitted the ETT‑small feature count—it should be d = 6, which will be corrected in the revised version. ).  Across our three real‑data benchmarks, d ranges from 2 to 6; based on the sensitivity analysis for K (Question 1), we set K = 3 in all real‑data experiments, since this choice consistently yielded stable accuracy and healthy expert utilization without excessive computational cost. All other architectural and training hyperparameters (e.g. hidden sizes, learning rate, dropout) are described in Appendix B.2. We emphasize that our goal was to demonstrate a general deep‑learning FoFR framework using a relatively simple, unified setup; as a fully extensible architecture, FAME can easily be combined with more complex deep models to suit different datasets.
>
> + Why is the cross attention necessary when there is continuous attention?
>
>   From a theoretical perspective, continuous attention operates **within each function** over time, performing a continuous, kernelized weighting of its own trajectory to capture **local–global intra‑function structure** (cf. Eqs. (3)–(5)). Cross‑attention, in contrast, acts **across functions at each time t by aggregating the concurrent states of the *other* functions, thereby modeling inter‑functional dependencies** explicitly (cf. Eqs. (7)–(10) and Sec. 4.3). Under Assumption 4, cross‑attention enjoys a global Lipschitz bound $L_{\text{cross}}$ and, when combined with continuous attention and the decoder, yields an end‑to‑end Lipschitz, sampling‑invariant mapping (Theorem 4), so removing cross‑attention weakens the model’s ability to represent cross‑channel couplings.
>
>   From an experimental perspective, and in response to the reviewers’ comments, we will add three ablation variants on both generic (Cases 1–2) and real‑data tasks—(1) without bidirectional attention, (2) without MoE (single expert), and (3) without cross‑attention—and report the resulting MSEs side by side to quantify each component’s contribution. The results will be presented in the table below.
>
>   | Model               |    Case 1 (100)    |    Case 1 (200)     |    Case 1 (500)    |   Case 2  (100)    |    Case 2 (200)    | Case 2      (500)  |
>   | :------------------ | :----------------: | :-----------------: | :----------------: | :----------------: | :----------------: | :----------------: |
>   | FAME w/o Bi‑dir     |   0.1832± 0.0150   |   0.0812± 0.0066    |   0.0654± 0.0055   | **0.1530± 0.0125** |   0.0528± 0.0023   |   0.0355± 0.0016   |
>   | FAME w/o MoE        |   0.1870± 0.0161   |   0.0828± 0.0072    |   0.0663± 0.0062   |   0.1578± 0.0132   |   0.0538± 0.0023   |   0.0362± 0.0017   |
>   | FAME w/o Cross‑attn |   0.1902± 0.0158   |   0.0815± 0.0078    |   0.0668± 0.0064   |   0.1602± 0.0143   |   0.0544± 0.0026   |   0.0375± 0.0018   |
>   | **FAME**            | **0.1806± 0.0152** | **0.0783±  0.0064** | **0.0635± 0.0056** |   0.1532± 0.0120   | **0.0511± 0.0023** | **0.0342± 0.0017** |
>
>   | Model                    | Hawaii Ocean Salinity |    Hawaii Ocean Temp |   Human3.6M Walking |     Human3.6M Eating | Human3.6M Sitting down |   ETDataset Oil Temp |
>   | ------------------------ | --------------------: | -------------------: | ------------------: | -------------------: | ---------------------: | -------------------: |
>   | Basis Expansion (best)   |       0.0780 ± 0.0039 |      0.0014 ± 0.0001 |     0.0359 ± 0.0018 |      0.0484 ± 0.0024 |        0.0122 ± 0.0006 |      0.0365 ± 0.0018 |
>   | FPCA                     |       0.0865 ± 0.0043 |      0.0025 ± 0.0001 |     0.0373 ± 0.0019 |      0.0099 ± 0.0005 |        0.0121 ± 0.0007 |      0.0355 ± 0.0018 |
>   | Kernel Method            |       0.0754 ± 0.0038 |      0.0025 ± 0.0001 |     0.0373 ± 0.0019 |      0.0099 ± 0.0005 |        0.0121 ± 0.0006 |      0.0355 ± 0.0016 |
>   | Gaussian Process         |       0.0931 ± 0.0047 |      0.0022 ± 0.0001 |     0.0360 ± 0.0018 |      0.0075 ± 0.0004 |        0.0107 ± 0.0005 |      0.0380 ± 0.0019 |
>   | FNN                      |       0.0766 ± 0.0038 |      0.0020 ± 0.0001 |     0.0373 ± 0.0019 |      0.0099 ± 0.0005 |        0.0121 ± 0.0006 |      0.0355 ± 0.0018 |
>   | FAME w/o bidir attention |       0.0751 ± 0.0023 |    0.00124 ± 0.00004 |     0.0327 ± 0.0010 |     0.0035 ± 0.00010 |        0.0071 ± 0.0002 |      0.0264 ± 0.0008 |
>   | FAME w/o MoE             |       0.0759 ± 0.0023 |    0.00130 ± 0.00004 |     0.0332 ± 0.0010 |     0.0038 ± 0.00011 |       0.0075 ± 0.00023 |      0.0271 ± 0.0008 |
>   | FAME w/o cross‑attention |       0.0773 ± 0.0024 |    0.00145 ± 0.00005 |     0.0344 ± 0.0011 |     0.0044 ± 0.00013 |       0.0083 ± 0.00025 |     0.0286 ± 0.00086 |
>   | **FAME**                 |   **0.0748 ± 0.0022** | **0.0012 ± 0.00004** | **0.0325 ± 0.0010** | **0.0034 ± 0.00010** |   **0.0070 ± 0.00020** | **0.0262 ± 0.00080** |
>
>
>   As the table shows, compared to the full FAME model, removing cross‑attention degrades performance, highlighting its role in capturing inter‑functional dependencies.
>
> Finally, we again thank you for your careful reading and hope you will enjoy the revised manuscript.

---

> > ### Comment · Reviewer_Xe76 · 2025-08-07
> >
> > I appreciate the additional experiments and thoughtful responses to my concerns. I still think there is more analysis necessary to understand the limitations of the method, especially with regards to selecting hyperparameters. I will keep my score as is.

---

### Note · Authors · 2025-08-12

We thank the program chairs, area chairs, and reviewers for giving our paper the opportunity to be presented at NeurIPS, and for their careful reading, constructive feedback, and discussions, which have improved the quality of our work.

Overall, our paper received positive feedback in both the initial submission and rebuttal stages. As summarized by **Reviewer pq9z**, our work replaces discrete self-attention with an NCDE-based continuous attention mechanism, treating both inputs and outputs as genuine functions rather than large vectors. This enables principled modeling of functional data while avoiding discrete approximation limits. The mixture-of-experts router — one set of weights per function instead of per time step — captures inter-function heterogeneity without prohibitive computational costs, balancing flexibility with efficiency. Theoretical analysis and experiments validate our approach. The clarity and organization were noted by **Reviewers Xe76 and EVtp**, which we believe makes the ideas accessible to a wider audience. In the rebuttal, we addressed every comment point-by-point, adding supplementary experiments, theoretical explanations, and clarifications to address concerns. **Reviewer Xe76** recognized our rebuttal, expressed agreement with our clarifications, and confirmed that they would maintain their initial score (5). We will also include an expanded discussion on hyperparameters in the final version. **Reviewer pq9z** stated that our results and analyses fully resolved their concerns, noting that the extra evidence strengthened the case for our contributions and suggesting that a higher score would be appropriate.

There are still a few points that we kindly ask the AC to consider. **Reviewer EVtp** did not respond after our rebuttal, but we believe our detailed replies addressed their concerns and invite the AC to review them. **Reviewer gU2z** raised concerns about computational efficiency; as with the innovations summarized above, our data-driven design, unlike traditional approaches relying heavily on prior assumptions, may be somewhat slower, which is common for such models, and while efficiency was not our primary focus we provided clarifications showing that our method runs well within practical limits — about one hour for offline training and seconds for inference — so in practice, no extra considerations or adjustments are required.

We again thank all reviewers and the AC for their efforts and valuable input.

---

### Decision · Program_Chairs · 2025-09-17

**Decision:**

Accept (poster)

**Comment:**

This paper proposes a functional attention mechanism FAME to perform functional-on-functional regression where both covariates and responses are functional data. A combination of tools (neural controlled differential equations, mixture-of-experts and continuous attention mechanisms) are deployed for this complex problem that involves infinite-dimensional data. Numerical experiments based on synthetic data shows the advantage of FAME over various baselines. The theoretical analysis establishes existence, uniqueness and global Lipschitz stability for every differential block along with an end-to-end Lipschitz bound and generalization guarantees.
Overall, this is a strong and well-structured paper that addresses key challenges in functional data, including intra-functional continuity, inter-functional interactions, and feature heterogeneity.  One weakness in the numerical experiment is that the comparison did not include vanilla transformers with simpler schemes, so we do not know specific kind of attention scheme works. The paper received scores 5-5-5-3.  I recommend it for publication.